# Development of Observation-based Global Multi-layer Soil Moisture Products for 1970 to 2016

Yaoping Wang[1,2], Jiafu Mao[2,*], Mingzhou Jin[1,3], Forrest M. Hoffman[4], Xiaoying Shi[2], Stan D. Wullschleger[2], and Yongjiu Dai[5]

[1] Institute for a Secure and Sustainable Environment, University of Tennessee, Knoxville, TN, USA 37902

[2] Environmental Sciences Division and Climate Change Science Institute, Oak Ridge National Laboratory, Oak Ridge, TN, USA 37830

[3] Department of Industrial and Systems Engineering, University of Tennessee, Knoxville, USA 37996

[4] Computational Sciences and Engineering Division and Climate Change Science Institute, Oak Ridge National Laboratory, Oak Ridge, TN, USA 37830

[5] School of Atmospheric Sciences, Sun Yat-sen University, Guangzhou, China 519082

*Correspondence to:* Jiafu Mao (1 Bethel Valley Rd, Oak Ridge, TN 37830; *maoj@ornl.gov*)

**Abstract.** Soil moisture (SM) datasets are critical to understanding the global water, energy, and biogeochemical cycles and benefit extensive societal applications. However, individual sources of SM data (e.g., in situ and satellite observations, reanalysis, offline land surface model simulations, Earth system model [ESM] simulations) have source-specific limitations and biases related to the spatiotemporal continuity, resolutions, and modeling/retrieval assumptions. Here, we developed seven global, gap-free, long-term (1970–2016), multi-layer (0–10, 10–30, 30–50, and 50–100 cm) SM products at monthly 0.5° resolution (available at https://doi.org/10.6084/m9.figshare.13661312.v1) by synthesizing a wide range of SM datasets using three statistical methods (unweighted averaging, optimal linear combination, and emergent constraint). The merged products outperformed their source datasets when evaluated with in situ observations (mean bias from −0.044 to 0.033 $m^3/m^3$, root mean squared errors from 0.076 to 0.104 $m^3/m^3$, Pearson correlations from 0.35 to 0.67) and multiple gridded datasets that did not enter merging because of insufficient spatial, temporal, or soil layer coverage. Three of the new SM products, which were produced by applying any of the three merging methods onto the source datasets excluding the ESMs, had lower bias and root mean square errors and higher correlations than the ESM-dependent merged products. The ESM-independent products also showed a better ability to capture historical large-scale drought events than the ESM-dependent products. The merged products generally showed reasonable temporal homogeneity and physically plausible global sensitivities to observed meteorological factors, except that the ESM-dependent products underestimated the low-frequency temporal variability in SM, and over-estimated the high-frequency variability for the 50–100 cm depth. Based on these evaluation results, the three ESM-independent products were finally recommended for future applications because of their better performances than the ESM-dependent ones. Despite uncertainties in the raw SM datasets and fusion methods, these hybrid products create added value over existing SM datasets because of the performance improvement and harmonized spatial, temporal, and vertical coverages, and they provide a new foundation for scientific investigation and resource management.

## 1 Introduction

High-quality global soil moisture (SM) datasets benefit many applications, such as understanding drought changes and ecosystem dynamics (Green et al., 2019; Kumar et al., 2019), studying land-atmosphere feedbacks (Li et al., 2020a), benchmarking model capabilities (Loew et al., 2013), and initializing weather and climate forecast systems (Sospedra-Alfonso and Merryfield, 2018). The majority of SM products fall into five categories: in situ measurements, satellite observations, offline land surface model (LSM) simulations, reanalysis, and Earth system model (ESM) simulations. In situ measurements provide the most direct SM observations at the point scale but are too sparse to be interpolated to the global level (spatial autocorrelation dies ~300 km (Gruber et al., 2016)). Satellite-derived SM records only penetrate the top few centimeters of soil and contain errors and spatial gaps typically caused by factors such as a change in path, dense vegetation, frozen soil, water bodies, and radio frequency interference (Llamas et al., 2020; Wang et al., 2012). Although a long-term (1979–present) concatenated SM dataset was developed by merging data from multiple satellites, the merged product did not fill the spatial gaps that existed in the source satellite datasets (Dorigo et al., 2012; EODC, 2021). The SM in LSM simulations usually spans multiple soil layers, and has no spatial or temporal gaps, which is convenient for regional and global analysis (Gu et al., 2019); however, LSM simulations may contain considerable errors because of inadequacies in the model physics, parameterization, and drivers (Andresen et al., 2020). Reanalysis datasets assimilate observations into LSMs or coupled forecast systems that have LSMs as a component, and are gap-free. Direct assimilation of remote-sensing SM, which has been the practice for some recent reanalysis—such as ECMWF Reanalysis 5 (ERA5) (de Rosnay et al., 2013) and Global Land Evaporation Amsterdam Model (GLEAM) (Martens et al., 2017)—is likely to improve the performance relative to free-running LSMs. Still, many reanalyses do not directly assimilate the observational SM, such as the Japanese 55-Year Reanalysis (JRA55) (Kobayashi et al., 2015) and Modern-Era Retrospective Analysis for Research and Applications Version 2 (MERRA2) (McCarty et al., 2016). Also, the meteorological variables, especially precipitation, simulated by the atmosphere model of the coupled reanalysis system may be biased, leading to inaccurate SM estimates by the intrinsic LSM component (Balsamo et al., 2015). Fully coupled ESMs, such as those for the Coupled Model Intercomparison Project phases 5 and 6 (CMIP5 and CMIP6) (Eyring et al., 2016; Taylor et al., 2012), provide SM simulations for both historical and future periods. ESMs, however, share the same uncertainty sources for the SM estimates as the LSMs; moreover, the SM in ESM simulations have internal variability-related uncertainties induced by unrealistic initialization from the preindustrial conditions rather than the real world (Eyring et al., 2016; Taylor et al., 2012).

There is active development toward generating more accurate, gap-free SM datasets. Methods for filling the spatial gaps in satellite observations have been under investigation, but the resulting estimates either cover short time periods or target only parts of the globe (Llamas et al., 2020; Wang et al., 2012). One global multi-layer SM product was generated by upscaling in situ observations using machine learning and selected SM predictors; however, it only focused on 2000–2019 (O and Orth, 2020). Unlike current global reanalyses that directly assimilate satellite SM (Martens et al., 2017; de Rosnay et al., 2013), some studies merged in situ observations, or both in situ and satellite observations, with offline LSMs to improve accuracy

while retaining complete spatiotemporal coverage. Nevertheless, these efforts were mainly conducted on the regional scale using limited sets of LSMs (Wu et al., 2018; Zeng et al., 2016; Gruber et al., 2016).

Therefore, the need exists to develop global merged SM products that comprehensively combine the information from the latest in situ and satellite observations, offline LSMs, reanalysis, and ESMs using advanced fusion methods. Because of the incorporation of various quality-controlled observations in the merging process, the merged products would likely perform better than the SM in the original LSMs or ESMs, while keeping the benefits of being gap-free in space and having long temporal and multi–soil-layer coverage. The fusion of multiple LSMs, reanalysis, and ESMs also involves ensemble averaging, which may reduce the SM uncertainties from individual models by cancelling the model-specific errors (Giorgi and Mearns, 2002). This study presents a group of SM products derived using three merging methods: unweighted averaging, optimal linear combination (OLC), and emergent constraint (EC). Unweighted averaging assigns equal weight to all the source datasets and does not use in situ information (see Sect. 2.1 for explanation for the exclusion). The OLC is an ensemble weighting and rescaling algorithm that is optimal in the sense that the weighted average minimizes the mean squared difference with respect to the site-level observations (Bishop and Abramowitz, 2013). The OLC method was previously found to lead to improved performance in the merged product relative to the source datasets in terms of the global evapotranspiration and runoff (Hobeichi et al., 2018, 2019). The EC method is common for reducing uncertainty in future ESM simulations (Mystakidis et al., 2016; Padrón et al., 2019). This method first uses data from multiple ESMs to establish physically meaningful and statistically significant relationship between the constraint variables that have observations and a target variable that has no observations, and then uses the relationship and actual observations to derive a constrained target variable (Mystakidis et al., 2016; Padrón et al., 2019). Given the clear physical relationships between the SM and meteorological variables, we hypothesized in this study that the EC method can be applied to reduce forcing-related biases in offline LSMs and reanalysis, and to align the natural internal variability in ESMs with the real world. Seven new monthly multi-layer SM datasets at 0.5°×0.5° resolution for 1971–2016 were produced by implementing these merging algorithms onto different combinations of the mentioned raw SM estimates. The merged products with different setups were then systematically evaluated against in situ measurements that were reserved for validation, semi-independent gridded SM datasets, drought indices, and meteorological variables.

## 2 Methods and source datasets

### 2.1 Overview

Figure 1 shows the schematic of the merging procedure to create the seven SM products. The unweighted averaging and OLC (Hobeichi et al., 2018, 2019) methods were applied over the observational or observation-forced datasets (i.e., offline LSMs, reanalysis, satellite [ORS]). The unweighted averaging did not use any in situ observations, because the in situ observations were sparse (~1,400 stations compared with ~60,000 grids in a 0.5° gap-free dataset over the global land surface; Sect. 2.2). In unweighted averaging, the in situ observations can only influence the merged values in the time steps and grids that

coincided with the observations. Therefore, the inclusion of in situ observations would have little influence on the results of unweighted averaging. Also, to validate a merged time step and grid, an un-merged observation must be available at the same

time step and grid, which would be difficult to achieve in data-sparse situations. The OLC method used in situ observations to constrain the ORS datasets. The EC method (Mystakidis et al., 2016) was applied over the ORS, CMIP5, CMIP6, the combination of CMIP5 and CMIP6 (CMIP5+6), and the combination of ORS, CMIP5, and CMIP6 (ALL) datasets (Eyring et al., 2016; Taylor et al., 2012), using gridded global meteorological observations as constraints. The use of ESM simulations with the EC method, but not with the unweighted averaging or OLC methods, was because the latter two methods resulted in

very inadequate performances when applied on the ESM simulations in a preliminary analysis (results not shown). Because the ORS datasets do not have uniform temporal coverage (Tables A1–A3), the unweighted averaging only used the ORS datasets that cover 1970–2016. For the OLC method, the ORS datasets were grouped based on three time ranges (1970–2010, 1981–2010, and 1981–2016) that were selected to maximize the available ORS datasets in each time range. For each time range, the ORS datasets that fully cover the time range were merged with the OLC method; if an ORS dataset fully covers two

or three time ranges, it was used in all the covered time ranges (see the "used time period" in Tables A1–A3). Then, the merged results for all three time ranges were concatenated into a consistent dataset covering the whole target period following a previous method for concatenating the remote-sensing SM (Dorigo et al., 2017; Liu et al., 2011, 2012) (Sect. 2.9). The CMIP5 and CMIP6 datasets always cover 1970–2016; when they were used jointly with the ORS datasets to produce the EC ALL (i.e., including all the source datasets) product, they were subset to the same time ranges as the ORS datasets, separately

processed and concatenated (bottom of Figure 1). All the synthesized monthly SM datasets are at 0.5° resolution, cover 1970–2016, and contain four depths (0–10, 10–30, 30–50, and 50–100 cm). The following sections provide more details of the datasets, merging methods, and processing procedures.

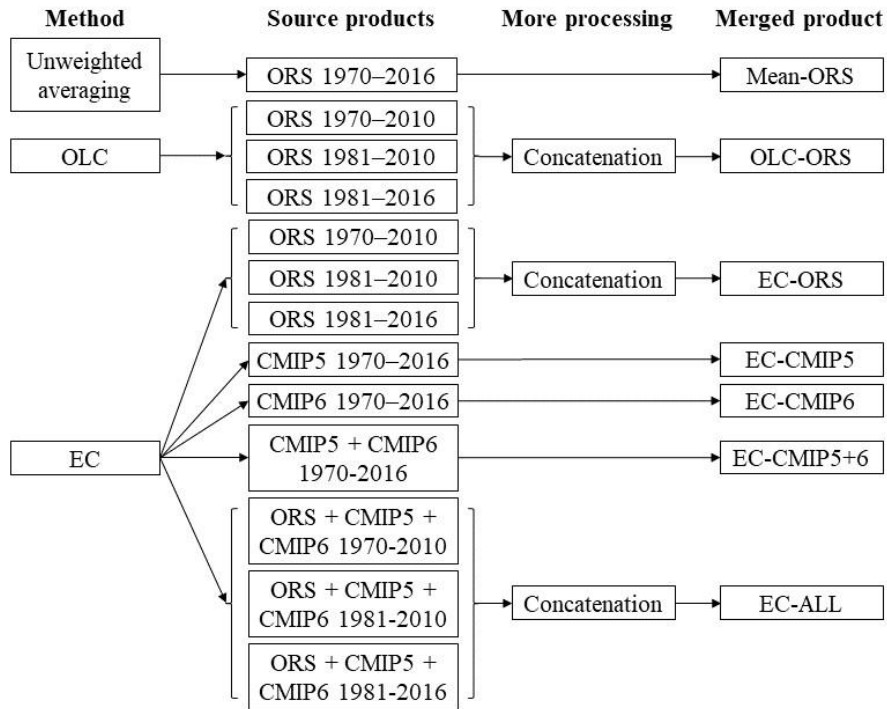

**Figure 1: Procedure of creating the merged SM products using different methods and source products.**

## 2.2 In situ SM observations

In situ SM observations were obtained from the International Soil Moisture Network (ISMN) (Dorigo et al., 2011, 2013). Only the observations associated with the ISMN quality flags "G" (good) or "M" (parameter value missing) were retained. The resulting dataset contains ~1,400 stations worldwide and spans from 1964 to the present. Because only a few stations were available at the beginning of the time period, only the observations in 1970 or later were used. To facilitate processing by the OLC method, the ISMN observations were aggregated to monthly 0.5° resolution and regular depths (0–10, 10–30, 30–50, and 50–100 cm). The aggregation to monthly resolution was simply averaging over all the available observations in each month at each station. Although it is desirable to apply a stricter criterion in the monthly averaging, such as treating a month as missing if observations exist fewer than 15 days in the month, applying such a criterion would exclude most of the stations in northern and eastern Asia, which only have 1–3 observations per month. Multiple methods were tested for the aggregation to 0.5° resolution: (1) simply averaging all the stations in each grid; (2) weighted-averaging the stations based on the percentage of grid area that the land cover of each station represents, using the Moderate Resolution Imaging Spectroradiometer (MODIS) MCD12C1 product (Friedl and Sulla-Menashe, 2015); and (3) the same as (2) except that if the total area of all the land covers that the stations represent does not account for a sufficient percentage of the grid area (40%), the grid was set to missing. Because all three methods resulted in similar performance in the merged products (results not shown), only the second method was adopted for the final products. The aggregation to regular depths was simply to average over all the available observations

in each depth interval. When an observation was taken on the interface of two depth intervals (e.g., exactly at 10 cm), the observation was assigned to the shallower depth interval (e.g., 0–10 cm). Figure A1 shows the aggregated ISMN observations at the 0.5° grid scale and the number of observations that falls into each land cover type. The observations are overrepresented in the developed part of the world; the most overrepresented land cover types include the deciduous broadleaf forests, grasslands, and croplands, whereas the most underrepresented land cover types are evergreen broadleaf forests, mixed forests, closed shrublands, and permanent wetlands. The number of available monthly observations did not decrease with deeper soil layers (global totals are 19,317 station-months for the 0–10 cm layer, 25,307 for the 10–30 cm layer, 25,011 for the 30–50 cm layer, and 20,660 for the 50–100 cm layer), although no observations were available for the closed shrublands and permanent wetlands across the deeper soil layers (10–30, 30–50, and 50–100 cm). After the ISMN observations were aggregated to monthly 0.5° resolutions, 60% of the month-grids were used as the observed SM values in the OLC method (the $o^{tj}$ variable; see Sect. 2.7), and the remainder were reserved for evaluating all the merged products. The training month-grids were uniformly randomly selected without distinguishing between space and time.

## 2.3 ORS, CMIP5, and CMIP6 SM products

Tables A1–A4 list the monthly gridded ORS, CMIP5, and CMIP6 results that were used as the SM source datasets for the merged products, as well as the depths (0–10, 10–30, 30–50, or 50–100 cm) and time ranges (1970–2016, 1970–2010, 1981–2010, or 1981–2016) for which the datasets were used. The name of the used SM variable was "mrlsl" in the CMIP5 collection and "mrsol" in the CMIP6 collection. Prior to merging, all the ORS, CMIP5, and CMIP6 datasets were bilinearly interpolated to 0.5° resolution, linearly interpolated to the target soil depths (0–10, 10–30, 30–50, or 50–100 cm), and masked with a common land mask using the National Center for Atmospheric Research (NCAR) Command Language 6.6.2 (UCAR/NCAR/CISL/TDD, 2019). Linear interpolation to all four soil depths could not be achieved for all the source datasets because the soil layers in some models are too shallow or too coarse. For example, the bottom depths of the Joint UK Land Environment Simulator (JULES) LSM are 10 cm, 35 cm, 1 m, and 3 m (Sebastian Lienert, 2019, Personal communication). Therefore, the 10–30 and 50–100 cm depths are contained within single layers in the JULES model and could not be directly interpolated. The four target soil depths here were selected to maximize the number of source datasets that could be interpolated to each depth. Although the Community Land Model version 4 (CLM4) simulates SM as deep as 421 cm, the model was only used for 0–10 and 10–30 cm (Table A3). This was because the SM values at deeper than 38 cm were very low (<0.0005 $m^3/m^3$ on global average), and even lower than the SM values at shallower than 38 cm in the same model (>0.0025 $m^3/m^3$ on global average). The common land mask was created by intersecting the grid points that satisfy two criteria: (1) all the datasets except the European Space Agency Climate Change Initiative (ESA CCI) v4.5, after being interpolated to 0.5°, have valid values; and (2) at least 50% of the land cover is not water bodies or permanent snow and ice in the MODIS MCD12C1 product (Friedl and Sulla-Menashe, 2015). Because of many spatial gaps, the ESA CCI v4.5 dataset was not used to create the common land mask and received special handling. In addition to being masked with the common land mask, the ESA CCI v4.5 dataset was masked using its accompanying quality flags, meaning at the time steps and grids that have snow coverage or temperature

below zero (flag = 1), dense vegetation (flag = 2), no valid SM estimates (flag = 4), SM values above physical boundary (flag = 8), or only unreliable SM values (flag = 16). The ESA CCI v4.5 dataset was excluded from merging at the time steps and grids in which the missing values exist.

## 2.4 Observed, CMIP5, and CMIP6 temperature and precipitation

The EC method, as implemented in this study (Sect. 2.8), requires the air temperature and precipitation forcings that correspond to each SM dataset, and observed air temperature and precipitation as inputs. The temperature and precipitation forcings that correspond to the ORS SM datasets are listed in Table A5. Because the ESA CCI v4.5 dataset is observational and the GLEAM v3.3a directly assimilates the ESA CCI dataset (Dorigo et al., 2017), these two datasets were assumed to correspond to the observed temperature and precipitation in the Climate Research Unit (CRU) TS v4.03 dataset (Harris et al., 2014). The temperature and precipitation forcings for the various reanalysis datasets were obtained from the same reanalysis. The temperature and precipitation forcings for the Multi-scale Synthesis and Terrestrial Model Intercomparison Project (MsTMIP) collection of LSMs were from the CRU NCEP v4 dataset, which, at monthly level, is equal to the CRU TS v3.20 dataset (Huntzinger et al., 2018). The temperature and precipitation forcings for the Trends and Drivers of the Regional Scale Sources and Sinks of Carbon Dioxide version 7 (TRENDY v7) collection of LSMs were from the CRU TS v3.26 dataset (Sitch and Friedlingstein, 2019, Personal Communication). The temperature and precipitation datasets that correspond to the CMIP5 and CMIP6 SM were from the same ESMs and ensemble members (Table A4). For observed air temperature and precipitation, the CRU TS v4.03 dataset (Harris et al., 2014) was used. All the temperature datasets were bilinearly interpolated, and the precipitation datasets conservatively interpolated, to 0.5° resolution using the NCAR Command Language 6.6.2 (UCAR/NCAR/CISL/TDD, 2019). All the temperature and precipitation datasets were limited to the same common land mask as the SM products.

## 2.5 In situ and gridded SM datasets for evaluation of the merged products

A recent discussion on the evaluation of coarse-scale soil moisture datasets noted that neither in situ observations, which have limited coverage and small spatial footprint, nor satellite and LSMs, which have retrieval or modeling errors, can be considered fully adequate for evaluation at the global scale; as such, a sound evaluation practice would require combining multiple sources of data (Gruber et al. 2020). Following this recommendation, the merged SM products were evaluated against the reserved 40% in situ observations (Sect. 2.2), as well as a few gridded reanalysis, satellite, and machine learning–upscaled SM datasets. Although the merging process aimed to use as many existing SM datasets as possible, the gridded datasets in Table 1 were not used in the merging because of incompatible vertical resolution, non-global spatial coverage, or short temporal coverage (Table 1). Such evaluation against multi-source gridded datasets complements the evaluation against in situ observations by providing sanity checks on the behavior of the merged products at large scales. All the evaluation datasets were bilinearly interpolated to 0.5°×0.5° and aggregated to a monthly level using the NCAR Command Language 6.6.2 (UCAR/NCAR/CISL/TDD, 2019). For the Soil Moisture and Ocean Salinity (SMOS) L3 dataset, which was available as monthly aggregates

(https://www.catds.fr/sipad/), only the data points with retrieval error (i.e., the "DQX" field) < 0.07 m³ m⁻³ were used, and the ascending (MIR_CLM4RA) and descending (MIR_CLM4RD) orbits were averaged. For the SMOS L4 dataset, which was only available at daily resolution, the days with a quality index of 1 (highest quality) were used, and the monthly averaging was restricted to the months with less than 13 missing days to be consistent with the downloaded monthly SMOS L3 dataset. The other evaluation datasets do not have gaps, so the aggregation to monthly level was straightforward. The SMOS L3 and L4 datasets were independent from the merged products. The SoMo.ml SM is mainly upscaled from the ISMN dataset (O and Orth, 2020) and therefore only semi-independent from the OLC ORS merged product, but independent from the unweighted averaging- or EC-based merged products. The GLEAM v3.3a 0–100 cm dataset is not independent from the ORS- or ALL-based merged products, which use the 0–10 cm part of the GLEAM v3.3a dataset (Table A2), but is independent from the CMIP5- or CMIP6-based merged products. The SMERGE v2 dataset uses the ESA CCI satellite data and therefore is non-independent from the ORS- or ALL-based merged products (Table A1), but is independent from the CMIP5- or CMIP6-based merged products.

The merged products were evaluated against the validation set of in situ observations and the gridded SM datasets using three common metrics: mean bias (Bias), root mean squared error (RMSE), and Pearson correlation coefficient (Corr). For evaluation against the in situ observations, the metrics were calculated both for the whole validation set and for each land cover type in consideration of the uneven distribution of ISMN observations across land cover types (Figure A1). The observational values used in each calculation were the land-cover–weighted averages (see Sect. 2.2), and the merged values were from the grids and time steps that have the observational values. For evaluation against the SMOS L3 gridded dataset, the 0–10 cm layer of the merged products and the source datasets (ORS, CMIP5, and CMIP6) were used. For evaluation against the other evaluation datasets, the merged and source datasets were linearly interpolated to depths of the evaluation datasets. The annual climatology, mean seasonal anomalies (i.e., the climatology of individual months minus the annual climatology), least-squares linear trends, and anomalies (i.e., the original values minus the mean seasonal cycle and trends) were calculated for each common grid cell and over the common time period between each pair of evaluated and evaluation datasets. Then, for each characteristic (climatology, seasonal cycle, linear trends, or anomalies), the Bias, RMSE, and Corr were calculated using the values of the characteristic pooled over all the common grid cells. When calculating the Bias, RMSE, and Corr for the trends, the insignificant trends at $p = 0.1$ were set to zero to prevent small random variability from influencing the results.

Table 1: Global and regional datasets that were compared against the merged SM products.

| Dataset | Type | Period | Depth (cm) | Resolution | Coverage | Reference |
|---------|------|--------|-----------|-----------|----------|-----------|
| SMOS L3 RE04 MIR_CLF3MA, MIR_CLF3MD | Satellite | 2010–2020 | Surface (0–5) | ~25 km | Global with missing values | (Al Bitar et al., 2017) |
| SMOS L4 SCIE MIR_CLM4RD | Reanalysis | 2010–2020 | 0–100 | ~25 km | Global with missing values | (Al Bitar et al., 2013) |
| GLEAM v3.3a | Reanalysis | 1980–2018 | 0–100 | 0.25° | Global | (Martens et al., 2017) |

| SMERGE v2 | Reanalysis | 1979–2019 | 0–40 | 0.125° | Contiguous United States | (Tobin et al., 2017) |
| SoMo.ml | Machine learning upscaled from in situ observations | 2000–2019 | 0–10, 10–30, 30–50 | 0.25° | Global | (O and Orth, 2020) |

## 2.6 Drought and meteorological datasets for evaluation of the merged products

In situ observations are sparse and represent much smaller spatial scales than the 0.5° grid of the merged products. Other global
and regional SM datasets have short temporal coverage and spatial gaps in the evaluation datasets, and non-independence
between the merged products and evaluation datasets. Given these limitations, and to further ascertain the quality of the merged
products, the new SM products were evaluated using process-based observational metrics, including the responses to
prominent historical drought events and historical climate change (e.g., precipitation, temperature, downwelling shortwave
radiation). The selected historical drought events were the United States drought of 1985–1992 and the Australian millennium
drought of 2002–2009 because of their macro-regional spatial scale and high severity; many other drought events would also
fit these criteria (Spinoni et al., 2019), but conducting a comprehensive assessment on drought events is beyond the scope of
this study. A Self-Calibrated Palmer Drought Severity Index (scPDSI) dataset (Dai et al., 2004) was used as the benchmark,
and the spatial patterns of SM anomalies and scPDSI were compared year-by-year during these two drought events. The
precipitation, temperature, and downwelling shortwave radiation datasets were from the Global Soil Wetness Project (GWSP)
version 3 reanalysis (Dirmeyer et al., 2006), which provide some independence from the CRU TS v4.03 temperature and
precipitation used in the EC method (Sect. 2.4). The SM climatic sensitivities were derived using the partial correlations with
each meteorological variable calculated conditional on the other two variables.

## 2.7 OLC method

Let $x_k^{tj}$ stand for the SM value of the source dataset $k$ ($k = 1, 2, …, K$) at time step $t$ ($t = 1, 2, …, T$) and grid $j$ ($j = 1, 2, …, S$),
and $o^{tj}$ be the observed SM values at time step $t$ and grid $j$ where $(t, j) \in V$ and $V$ is the subset of grids and time steps that
have observed SM. OLC calculates the final estimated SM ($\mu_e^{tj}$) as a weighted average, with $w_k$ denoting the weight of source
dataset $k$, using Eq. (1):

$$\mu_e^{tj} = \sum_{k=1}^{K} w_k x_k^{tj} \tag{1}$$

The optimal vector of weights for the source datasets, $\boldsymbol{w} = [w_1, w_2, …, w_K]^T$, which minimizes the mean squared error subject
to $\sum_{k=1}^{K} w_k = 1$, is a function of the error covariance matrix of the source datasets ($\boldsymbol{A}$) following Eq. (2):

$$\boldsymbol{w} = \frac{\boldsymbol{A}^{-1}\boldsymbol{1}}{\boldsymbol{1}^T \boldsymbol{A}^{-1}\boldsymbol{1}} \tag{2}$$

The OLC procedure without a constant term requires the source datasets to be unbiased (Bishop and Abramowitz, 2013), but
the ORS datasets are biased relative to the in situ observations. Therefore, to prevent the biases from influencing the weights,

the error covariance matrix $A$ was the covariance matrix of locally centered errors ($e_k^{tj}$), which were calculated following Eq. (3):

$$e_k^{tj} = \left(x_k^{tj} - x_k^{\cdot j}\right) - \left(o^{tj} - o^{\cdot j}\right) \quad , \quad (t,j) \in V \tag{3}$$

where $x_k^{\cdot j}$ is the time-averaged SM value of the source dataset $k$ at grid $j$, and $o^{\cdot j}$ is the time-averaged observed SM value at grid $j$. The time averaging was over the time steps in which observed SM values exist in each grid $j$. The grids and time steps for which the centered errors exist were pooled together to create a single vector of errors for each source dataset. This vector of errors $e_k^{tj}$ was then used to calculate the error covariance matrix $A$. Therefore, the derived weights were optimal with regard to the locally centered errors, not the un-centered errors $\left(x_k^{tj} - o^{tj}\right)$. However, this limitation had to be accepted because the ISMN observations were too sparse to enable estimating the biases at every grid in space, and a constant bias could not be assumed over the large spatial domain of the study, which spans very dry to very wet climate zones.

The OLC procedure was implemented in Python 3.6.3 under a CentOS Linux environment. Different functions for calculating the error covariance matrix in Python were compared to eliminate potential numerical instability in matrix inversion. The results were found to be similar, but the ShrunkCovariance function in Scikit-learn v0.21.3 (Pedregosa et al., 2011) generated slightly better validation performance for the estimated SM than the other tested functions. Therefore, ShrunkCovariance was selected for calculating $A$.

In addition to estimating SM, the OLC procedure also calculates the associated uncertainty in the form of standard deviation ($\sigma_e^{tj}$) based on adjusted weights ($\widetilde{w}_k$), adjusted source SM datasets ($\tilde{x}_k^{tj}$), and the estimated SM ($\mu_e^{tj}$), following Eq. (4):

$$\sigma_e^{tj} = \sqrt{\sum_{k=1}^{K} \widetilde{w}_k \left(\tilde{x}_k^{tj} - \mu_e^{tj}\right)^2} \tag{4}$$

The adjusted weights are a function of the original weights ($w_k$) and a parameter $\alpha$ to have $\widetilde{w}_k \geq 0$ and maintain $\sum_{k=1}^{K} \widetilde{w}_k = 1$, following Eqs. (5) and (6):

$$\widetilde{w}_k = \frac{w_k + (\alpha - 1)\frac{1}{K}}{\alpha} \tag{5}$$

$$\alpha = \begin{cases} 1 & \text{if all } w_k \text{ are nonnegative} \\ 1 - K \min(w_k) & \text{if the minimum } w_k \text{ is negative} \end{cases} \tag{6}$$

The adjusted source SM datasets ($\tilde{x}_k^{tj}$) are linear functions of the source SM datasets ($x_k^{tj}$) and the estimated SM ($\mu_e^{tj}$) through parameters $\alpha$ and $\beta$, where the parameter $\beta$ is a function of the discrepancy between the observations and the estimated SM ($s_e^2$), following Eqs. (7)–(9):

$$\tilde{x}_k^{tj} = \mu_e^{tj} + \beta\left(x_k^{\cdot j} + \alpha\left(x_k^{tj} - x_k^{\cdot j}\right) - \mu_e^{tj}\right) \tag{7}$$

$$\beta = \sqrt{\frac{s_e^2}{\frac{1}{N}\sum_{j,t\in V}\sum_{k=1}^{K}\widetilde{w}_k\left(x_k^j + \alpha\left(x_k^{tj} - x_k^j\right) - \mu_e^{tj}\right)^2}} \tag{8}$$

$$s_e^2 = \frac{\sum_{j,t\in V}\left(\mu_e^{tj} - o^{tj}\right)^2}{N - 1} \tag{9}$$

where $V$ is the subset of grid and time step combinations that have observed SM, and $N$ is the total number of grid-time steps that have observed SM (i.e., $N = |V|$).

## 2.8 EC method

For establishing the EC relationship, temperature and precipitation were selected to be the constraint variables because of their significant roles in controlling evapotranspiration from and recharge to the soil water. SM was the target variable. For each target year, month, grid, and soil depth, a linear regression relationship was fitted using SM anomalies as the predictand, and temperature and precipitation anomalies as the predictors. The SM, temperature, and precipitation anomalies of each source dataset (i.e., the datasets in the ORS, CMIP5, CMIP6, CMIP5+6, or ALL group; Figure 1) were calculated by removing the monthly climatology of 1981–2010. The vectors of SM, temperature, and precipitation anomalies in each regression relationship consisted of the anomalies for the target year, month, and soil depth over the 9 nearest grids to the target grid and over all the source datasets. If the fitted regression slopes of both temperature and precipitation anomalies were significant at $p = 0.05$, the fitted regression was used as the EC relationship. If either slope was insignificant, the regression was refitted using only precipitation or temperature anomalies as the predictor. If the refitted slope of precipitation (temperature) anomalies was significant at $p = 0.05$ and had a lower $p$-value than the refitted slope of temperature (precipitation) anomalies, the refitted regression with precipitation (temperature) anomalies was used as the EC relationship. If neither of the refitted slopes was significant at $p = 0.05$, the EC relationship was deemed insignificant for this year, month, grid, and soil depth. After the significant EC relationships were obtained, the observed temperature and precipitation were converted to anomalies relative to the monthly climatology of 1981–2010, and fed into the EC relationships to generate constrained SM anomalies. Finally, the constrained SM anomalies were added to the mean monthly climatology over all the source datasets to generate constrained SM values. For the combinations of year, month, grid, and soil depth that did not have significant EC relationships, the mean monthly climatology over all the source datasets was used as the constrained SM values. Uncertainties in the EC-constrained SM values were estimated using standard deviation of the prediction of the linear regressions, calculated by the "wls_prediction_std" function of the Python package statsmodels (Seabold and Perktold, 2010). Uncertainties in which there were no significant EC relationships were estimated using the standard deviation of the source datasets.

The fitted EC relationships are summarized in Figures A3 and A4 using the average values of the significant regression coefficients, and the percentage of significant regression coefficients for temperature and precipitation, respectively. The regression coefficients for temperature were mostly negative, and for precipitation mostly positive, which can be explained by

the fact that higher temperature causes higher evaporative loss of water from soil, and higher precipitation causes more recharge of water into soil. In the Sahara region, the average regression coefficients were mostly positive for temperature (Figure A3), which might be related to interannual correlation between precipitation and temperature caused by the West African monsoon (Zhang and Cook, 2014). For the ORS datasets at 30–50 and 50–100 cm, the regression coefficients were also mostly positive for temperature (Figure A3). The ORS datasets at these depths only represent three sets of meteorological forcings (GLDAS

NOAH025_M2.0, CRU TS v3.20 for MsTMIP, and CRU TS v3.26 for TRENDY v7; Tables A1–A3 and A5). Therefore, the EC relationships for the ORS datasets at these depths likely have high uncertainty. Because only small percentages of EC relationships were significant at these depths (Figures A3 and A4), the counterintuitive regression coefficients were unlikely to have a large impact on the merged product. In general, the percentages of significant EC relationships were higher for the CMIP5, CMIP6, CMIP5+6, and ALL datasets than for the ORS datasets, suggesting that more diverse source datasets lead to

stronger EC relationships (Figures A3 and A4). In the preliminary analysis, additional setups for the regression were tested, including whether the regression should use the actual values of SM, temperature, and precipitation or the anomalies, and whether the regression should use only the target grid or the 9 nearest grids. The setup with anomalies and the 9 nearest grids was found to result in more significant regression coefficients and better performance (results not shown).

## 2.9 Temporal concatenation and homogeneity test against in situ and gridded SM datasets

Because some ORS datasets were not available for the entire 1970–2016 period, separate OLC ORS, EC ORS, and EC ALL datasets were produced for three different periods (1970–2010, 1981–2016, and 1981–2010) and were concatenated into a continuous 1970–2016 product using a previous intercalibration approach (Figure 1) (Dorigo et al., 2017; Liu et al., 2011, 2012). To perform the concatenation on the estimated SM values, the merged product for each of the three periods was decomposed into monthly climatology and monthly anomalies, with the monthly climatology being calculated on the

overlapping period (1981–2010). Then, the anomalies of the 1970–2010 and 1981–2010 product were rescaled to have the same cumulative distribution function (CDF) as the anomalies of the 1981–2016 product during their overlapping period (1981–2010) using the piecewise linear CDF matching technique (Liu et al., 2011). In the first step of the CDF matching technique, the 0th, 5th, 10th, 20th, 30th, 40th, 50th, 60th, 70th, 80th, 90th, 95th, and 100th percentiles of the anomalies of each product during the overlapping period were identified on their CDF curves. In the second step, the percentiles of the 1970–

2010 and 1981–2010 products were plotted against the percentiles of the 1981–2016 product. A linear line was drawn between each two adjacent percentiles (e.g., the 5th and 10th percentiles), resulting in 12 linear segments. In the last step, the anomalies of the 1970–2010 and 1981–2010 datasets that fell into each interval of percentiles (e.g., 5th–10th) were rescaled using the equations of the linear segments. Values outside the range of the monthly anomalies during the overlapping period were rescaled using the equation of the closest linear segment. A graphic illustration of the CDF matching technique can be found

in Fig. 3 of Liu et al. (2011). The CDF matching was conducted for all the months as a whole, rather than for each month separately, because the latter setup would result in too few data points (36 data points during 1981–2016) to robustly determine the percentiles. The rescaled anomalies were added back to the monthly climatology of each product. Finally, the three added-

back products were concatenated by using the 1970–2010 product for 1970–1980, the 1981–2010 product for 1981–2010, and the 1981–2016 product for 2010–2016. To concatenate the estimated SM uncertainty of the OLC method ($\sigma_e^{tj}$; see Sect. 2.7) and the EC method (see Sect. 2.8), the uncertainty values of 1970–2010 and 1981–2010 were directly rescaled to the 1981–2016 values using the CDF matching without prior conversion to monthly anomalies, and the rescaled uncertainty values concatenated like the mean values.

Despite the intercalibration procedure, temporal discontinuity may still exist in the OLC ORS, EC ORS, and EC ALL products because their source datasets were different in the 1970–1980, 1981–2010, and 1981–2016 periods. To test this possibility, a previously demonstrated homogeneity test procedure (Su et al., 2016) was applied to determine whether statistically significant discontinuities in mean or variance exist between the 1970–1980 and 1981–2010 periods, or between the 1981–2010 and 1981–2016 periods. The procedure involves calculating $Q$-values between the time series of interest, $Y$, and one or multiple reference time series, $X_i$, $(i = 1,2,..,K)$, that are in the same grid as $Y$, using Eq. (10):

$$Q = Y - \frac{\sum_{i=1}^{K} V_i (\beta_i X_i - c_i)}{\sum_{i=1}^{K} V_i} \tag{10}$$

where $\beta_i$ and $c_i$ are the linear regression coefficients between $X_i$ and $Y$, and the weighting coefficient, $V_i$, is equal to the square of $\beta_i$ if $\beta_i$ is positive, and zero if $\beta_i$ is negative. Then, the procedure uses the Wilcoxon rank-sum test to determine if the mean values of $Q$ are significantly different between two time periods, and the Fligner-Killeen test to determine if the variances of $Q$ are significantly different between two time periods. Like in the original study (Su et al., 2016), a time series $X_i$ was only selected for comparison if the Pearson correlation between $X_i$ and $Y$ was greater than 0.8 and significant at a $p$-value of 0.01 or smaller, and if at least three $Q$-values could be calculated in each of the two compared time periods. In this study, the $Y$ time series were the time series in each grid in a merged product. Two types of time series $X_i$ were used. One was the time series of original ISMN observations (i.e., not interpolated to 0−10, 10−30, 30−50, or 50−100 cm, aggregated to grid level, or split into validation or training sets) that were located in the same grid (Sect. 2.2). The other type was the time series of the gridded SM datasets for evaluation (Sect. 2.5) in the same grid. For the latter type, the merged products were linearly interpolated to the same depths as the gridded SM datasets.

## 3 Results

### 3.1 Evaluation against the validation set of ISMN SM observations

When evaluated on the whole validation set, the Bias of the merged products ranged from −0.044 to 0.033, the RMSE ranged from 0.076 m³/m³ to 0.104 m³/m³, and the Corr ranged from 0.35 to 0.67, across the four soil depths (Figure 2). The merged products generally showed smaller magnitude of Bias, smaller RMSE, and higher Corr than the source datasets from which the products were merged (Figure 2). The Bias of both the source and merged datasets shifted from mostly positive to mostly negative from the shallowest to the deepest soil layer, indicating a tendency toward the overestimation of vertical SM gradient;

the shallower soil layers also tended to have lower RMSE and higher Corr than the deeper layers (Figure 2). The Bias values of individual merged products were similar; the RMSE and Corr values of the ORS-based merged products (Mean ORS, OLC ORS, EC ORS) were better than the EC ALL product; and the RMSE and Corr values of the EC ALL product were better than the CMIP5- or CMIP6-based merged products (EC CMIP5, EC CMIP6, EC CMIP5+6) (Figure 2).

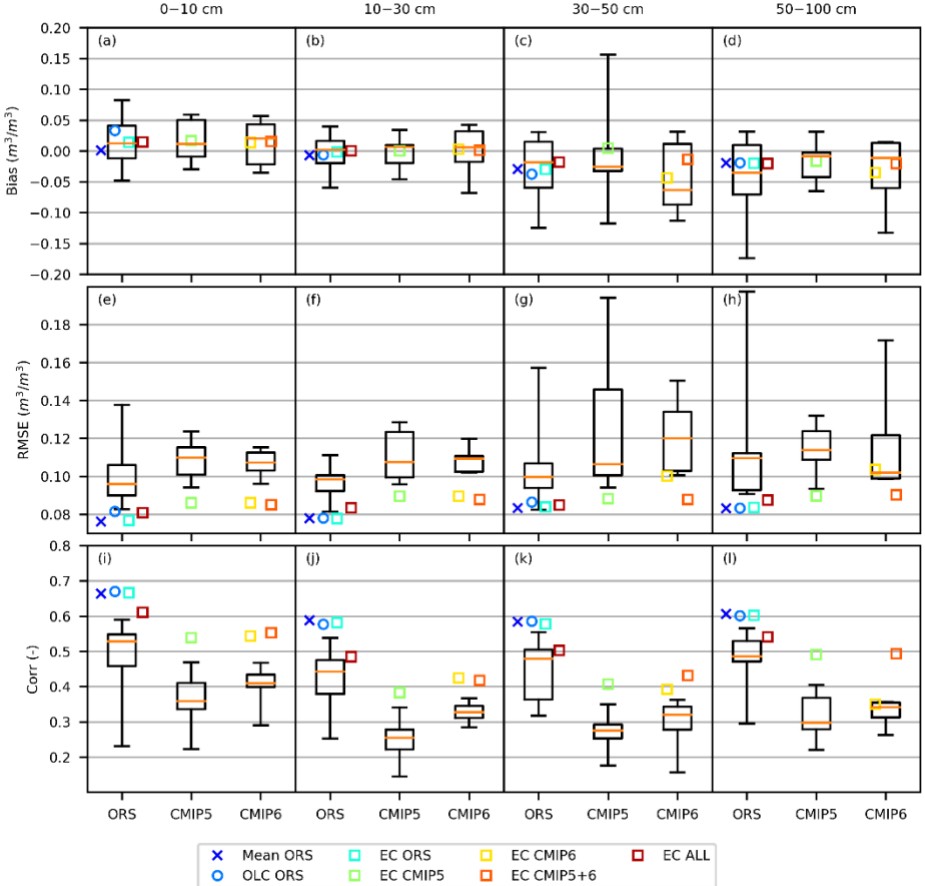

**Figure 2: Performance of the original ORS, CMIP5, and CMIP6 datasets (boxplots) and the merged products (scatter plots) on the validation set of observations. The boxplots show (from top to bottom) maximum, 75th percentile, median, 25th percentile, and minimum. The ORS boxplot includes all the ORS datasets evaluated on their available years.**

The merged products showed lower magnitude of Bias, lower RMSE, and higher Corr than the source datasets across most of the land cover types (Figure A5). The exceptions were the shallower (0–10 and 10–30 cm) soil layers over the water bodies, evergreen needleleaf forests, and evergreen broadleaf forests, where the merged products produced similar RMSE to the bulk of the source datasets (Figure A5e–h), and the deeper (30–50 and/or 50–100 cm) soil layers over the open shrublands, urban and built-up lands, cropland/natural vegetation mosaics, and barren lands, where the merged products showed similar or even lower Corr, and similar RMSE to the bulk of the source datasets (Figure A5e–i). Although the merged datasets considerably overestimated the SM in the water bodies and evergreen needleleaf forests (Bias = 0.016 to 0.146 $m^3/m^3$; Figure A5a–d), the

high Corr for these two land cover types (0.30 to 0.72 for the ORS-based merged products, −0.27 to 0.55 for the other merged products [Figure A5i–l]) indicated good ability to track the spatio-temporal variability. The hybrid products underestimated the SM in the 0–10 cm layer of the evergreen broadleaf forests (Bias = −0.174 to −0.095 $m^3/m^3$), the deciduous needleleaf forests (Bias = −0.162 to −0.055 $m^3/m^3$), and the deeper soil layers of many other land cover types (Figure A5a–d). The Corr values of the merged products were also very low in the evergreen broadleaf forests (−0.81 to 0.05; Figure A5i–l). Similar to at the global level, the ORS-based merged SM tended to outperform the EC ALL and the CMIP5- and CMIP6-based merged products (Figure A5). The three merging methods performed similarly over most land cover types, but the OLC method (OLC ORS product) had lower RMSE and higher Corr than the other two methods (Mean ORS and EC ORS products) over the urban and built-up lands, crop/natural vegetation mosaic, and barren land cover types (Figure A5).

## 3.2 Evaluation against global and regional gridded SM datasets

The evaluating results for the merged SM products against independent or semi-independent, gridded SM datasets were highly dependent on the evaluation dataset. For example, as shown above each panel in Figure 3, the maximum Bias, RMSE, and Corr among the merged and source datasets in the evaluation of climatology against the SMOS L4 0–100 cm dataset were, respectively, 0.12 $m^3/m^3$, 0.16 $m^3/m^3$, and 0.16, whereas the same metrics were −0.02 $m^3/m^3$, 0.11 $m^3/m^3$, and 0.81 in the evaluation against the GLEAM v3.3a 0–100 cm dataset. To emphasize the differences between the merged and the source datasets, rather than across the evaluation datasets, Figure 3 displays the evaluation metrics in normalized units, with the maximum value across the merged and source datasets of each matric set to 100%. Within each column of each panel of Figure 3, the merged products generally had lower RMSE and higher Corr than the average RMSE and Corr of corresponding source datasets, and the ORS-based products generally had lower RMSE and higher Corr than the CMIP5- or CMIP6-based products. The magnitudes of Bias were often similar between the merged and the source datasets for climatology, trend, and anomalies, but the Bias in seasonality of the merged datasets was generally better than that of the source datasets.

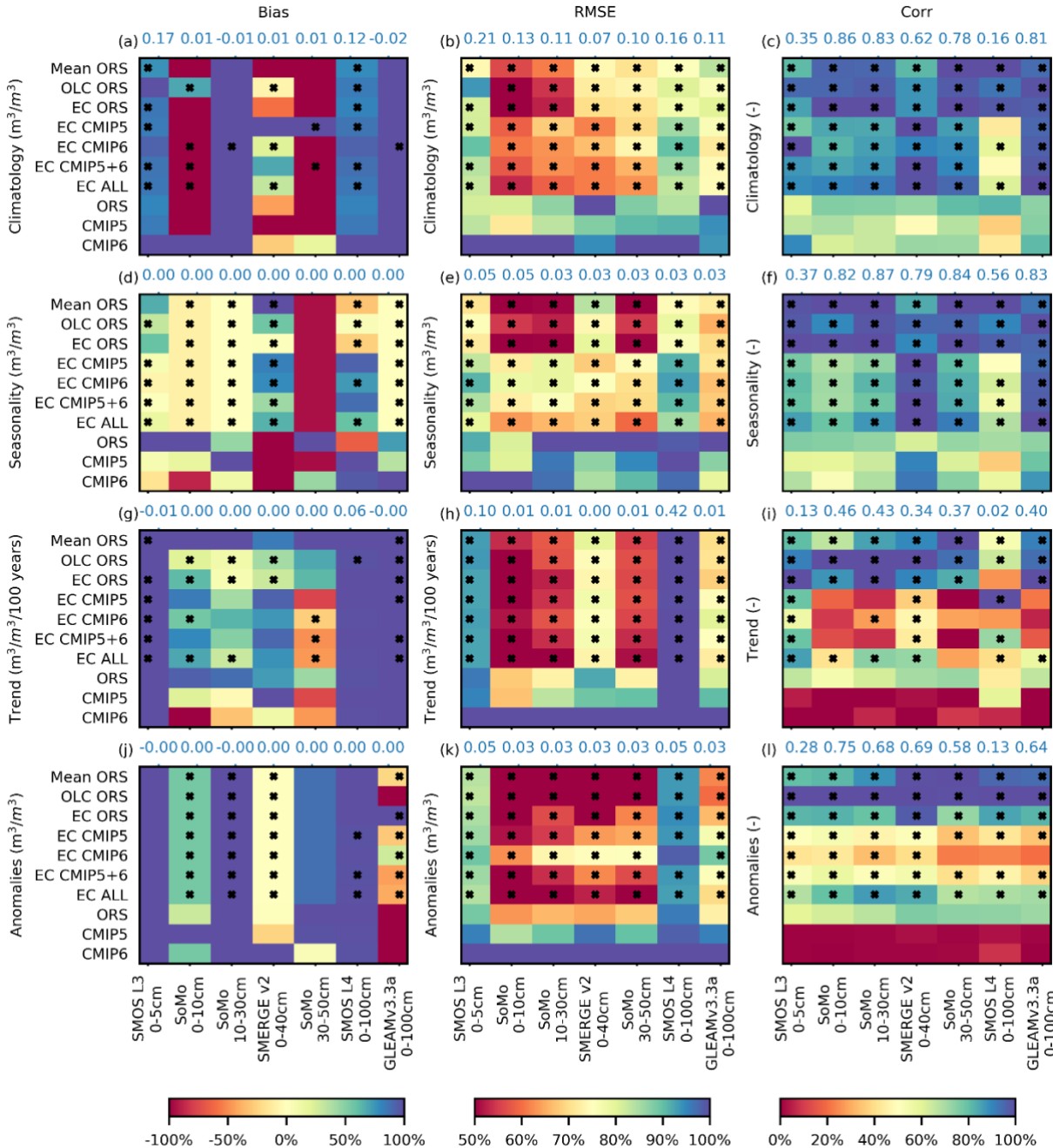

**Figure 3: The normalized Bias, RMSE, and Corr among the annual climatology, seasonal cycle, linear trends, and anomalies of the individual merged products (Mean ORS through EC ALL) and source datasets (ORS, CMIP5, CMIP6), and the global and regional datasets for evaluation (SMOS L3, SoMo, SMERGE v2, SMOS L4, GLEAMv 3.3a). The normalization involves dividing each value by the column-wise maximum in each panel and multiplying by 100%, and was performed to prevent all the values of each column from showing the same color. The blue number at the top of each column is the column-wise maximum to the precision of two decimal points. The asterisk (\*) indicates that the magnitude of the Bias, RMSE, or Corr of the merged product is better than the**

 **product's source datasets in the same column (for EC CMIP5+6, the comparison was made against the average of CMIP5 and CMIP6; for ALL, against the average of ORS, CMIP5, and CMIP6). The displayed evaluation metrics of the ORS, CMIP5, and CMIP6 were the average value over the individual source datasets in each group.**

### 3.3 Homogeneity test against in situ and gridded SM datasets

Because of the screening criteria (Sect. 2.9), few in situ observational time series were available for conducting the homogeneity test. Therefore, homogeneity tests using in situ SM data were only performed for between 1 and 10 grids (exact numbers not shown) for the various combinations of time periods (between 1970–1980 and 1981–2010, or between 1981–2010 and 1981–2016) and depths (0−10, 10−30, 30−50, and 50−100 cm). None of these test results showed significant discontinuity, but the scarcity of tested grids rendered the finding inconclusive.

With the gridded SM datasets, the majority of the global grids satisfied the screening criteria. The left two columns of Figure 6 show that for all the merged products, no significant discontinuity in mean existed between the time periods 1970−1980 and 1981−2010. The right two columns of Figure 6 show that discontinuity in variance existed between the time periods 1970−1980 and 1981−2010, but the percentages of discontinuous grids were similar between the concatenated products (OLC ORS, EC ORS, EC ALL) and the other products that were based on the same source datasets throughout 1970−2016 (Mean ORS, EC CMIP5, EC CMIP6, and EC CMIP5+6). Although the fitting of a separate regression for each year and month in the EC procedure (Sect. 2.8) might have introduced inhomogeneities into to the EC-based products, the unweighted averaging method behind Mean ORS did not have the same concern. However, the Mean ORS product had the highest percentages of discontinuous grids among all the merged products. These results indicate that the identified discontinuities were more likely caused by systematic differences between the evaluation datasets (SMERGE v2 and GLEAM v3.3a) and the source datasets for merging (ORS, CMIP5, and CMIP6), rather than the concatenation procedure. The homogeneity test between 1981−2010 and 2011−2016 had similar results to the test between 1970–1980 and 1981–2010. Virtually no discontinuities in mean were identified, and similar percentages of discontinuous grids were identified in the concatenated products (OLC ORS, EC ORS, EC ALL) and the others (Figures A6−A7).

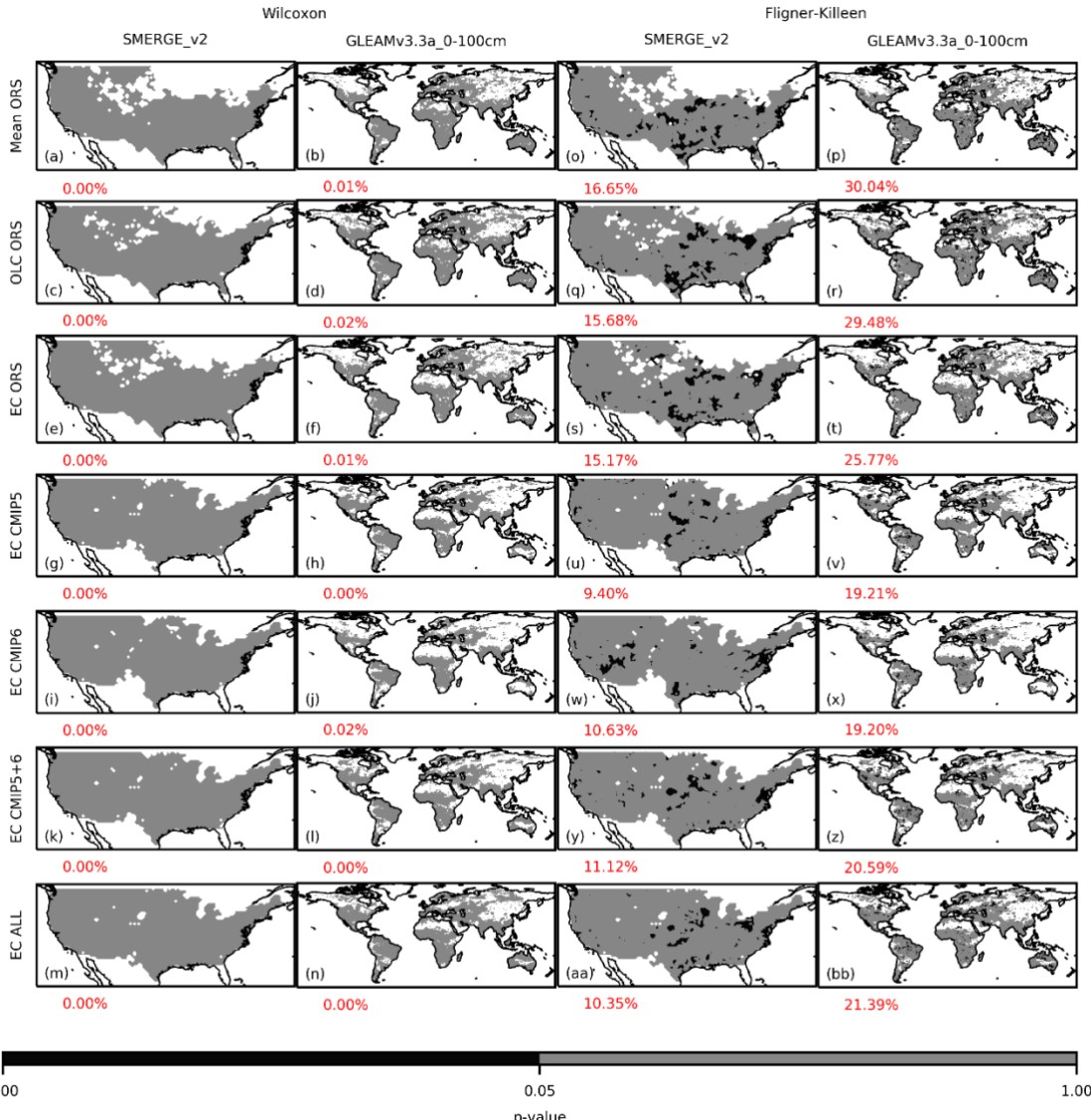

**Figure 4: The *p*-values of the homogeneity test on the discontinuity in mean (via Wilcoxon rank-sum test), and in variance (via Fligner-Killeen test) between the time periods 1970−1980 and 1981−2010, in the merged products. The SoMo datasets could not be used to evaluate discontinuity between these two time periods because SoMo only spans 2000 to 2019. The red numbers beneath each panel indicate the percentage of grids that had significant discontinuity (*p* ≤ 0.05) in the panel. Blank grids exist because the SMERGE v2 or GLEAMv3.3a 0−100 cm data in these grids did not satisfy the screening criteria for the homogeneity test.**

## 3.4 Evaluation against selected drought events

Lower values in scPDSI and SM anomalies are indicative of drier conditions, and higher values indicate wetter conditions. For the United States 1985–1992 drought, the scPDSI, 0−10 cm SM anomalies, and 0−100 cm SM anomalies all showed gradual expansion of drought from 1985 to 1988, and gradual alleviation from 1989 to 1992, with the most severe drought being

reached in the northern Great Plains in 1988 (Figures 4 and A9). For the Australia 2002–2009 drought, the ORS-based 0−10 and 0−100 cm SM anomalies captured the pan-Australian drought shown by the scPDSI in 2002−2003, 2005, and 2007−2009, and the eastern-Australian drought in 2004 and 2006 (Figures A8 and A10). The CMIP5- and CMIP6-based SM anomalies also mostly captured the Australian drought patterns but did not capture the pan-Australian drought in 2007 and 2008 (Figures A8 and A10).

To better quantify the similarity between the scPDSI and SM anomalies, Spearman correlations (Hollander et al., 2013) were calculated and are shown above each panel in Figures 4 and A8−A10. The Spearman correlation metric was deemed suitable for measuring the similarity because the magnitudes of scPDSI, which is a unitless standardized index, and of SM anomalies $(m^3/m^3)$, are not comparable. Spearman correlation is not sensitive to magnitudes because the metric is calculated using the rank of each *x*-value among all the *x*-values, and the rank of each *y*-value among all the *y*-values, for an *x-y* pair of time series (Hollander et al., 2013). The Spearman correlations between scPDSI and the ORS-based SM anomalies were between 0.698 and 0.890 for the United States (Figures 4 and A9), and 0.427 to 0.872 for Australia (Figures A8 and A10). For the purely CMIP5- or CMIP6-based products, the Spearman correlations were between −0.147 and 0.850 for the United States (Figures 4 and A9), and 0.005 to 0.872 for Australia (Figures A8 and A10). The EC ALL product, which combines ORS, CMIP5, and CMIP6 source datasets, had Spearman correlations that tended to be in the middle of the ORS- and the CMIP5/CMIP6-based products. The better performances of the ORS-based than the CMIP5- and CMIP6-based merged products were consistent with the evaluation results against in situ observations (Figures 2 and A5).

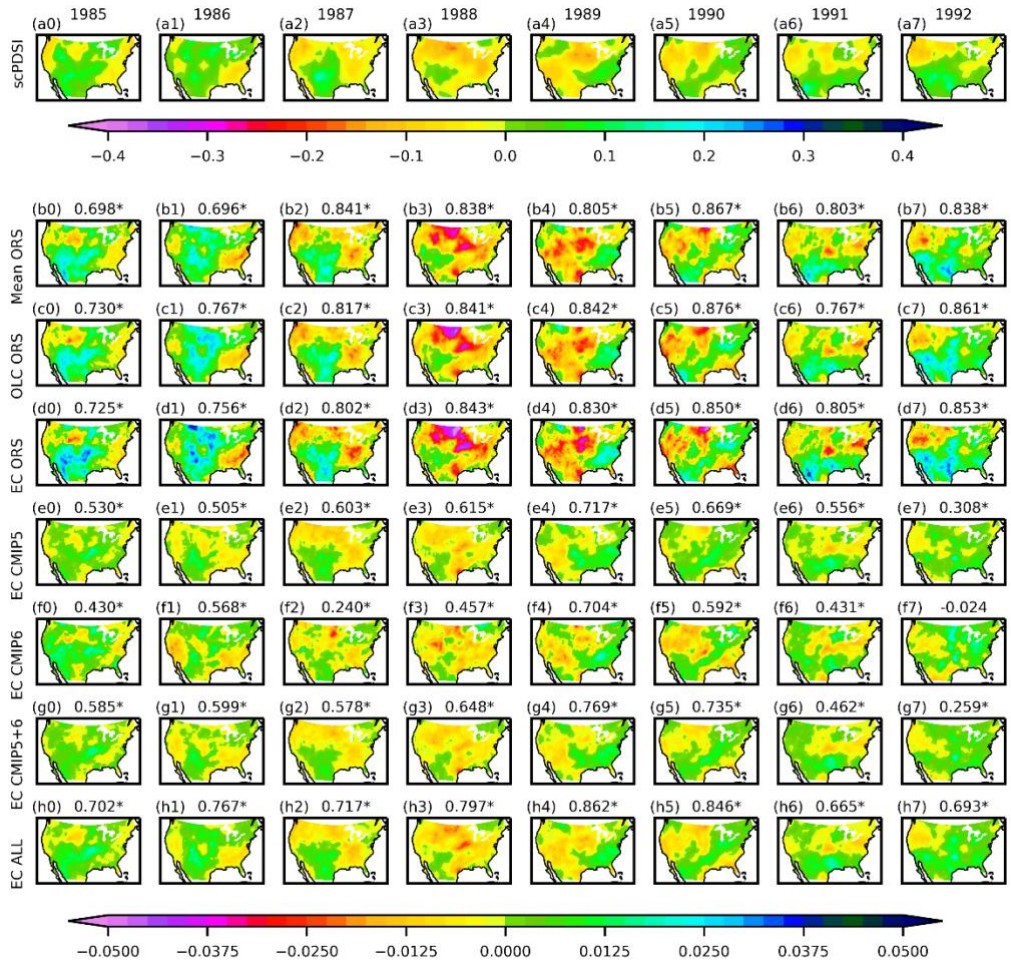

**Figure 5: The annual mean scPDSI anomalies (no unit) and annual mean SM anomalies (m³/m³) during the US drought 1985–1992. The numbers above the plots are the Spearman correlation between the anomalies of the merged product and scPDSI, and the asterisk (*) indicates that the correlation is significant at $p = 0.05$. The anomalies were calculated relative to the climatology of 1970–2016. The SM anomalies are for 0–10 cm.**

**3.5 The spatial and temporal characteristics of the merged SM products**

Because all the merging procedures essentially involve averaging over multiple source datasets (see Sects. 2.1, 2.7, and 2.8), the variability and trends of the merged products may be damped compared with the source datasets because of mutual cancellation. To ascertain whether this is the case, power spectral densities were calculated on regionally averaged time series of the merged products and the source datasets (i.e., ORS, CMIP5, and CMIP6), and were compared in Figures 6 and A10. The regions used were the Intergovernmental Panel on Climate Change (IPCC) Special Report on Managing the Risks of Extreme Events and Disasters to Advance Climate Change Adaptation (SREX) regions (Field et al., 2012; Figure A11). The power spectral densities of the Mean ORS, OLC ORS, and EC ORS products very rarely exceeded the boundaries of the source

datasets (e.g., panel c2 of Figure 6, panel j0 of Figure A12), showing that at least at a regional level, these merged products did not underestimate the temporal variability in SM. The variabilities of the EC CMIP5, EC CMIP6, EC CMIP5+6, and EC ALL products were generally within the boundaries of the source datasets at the 0−10, 10−30, and 30−50 cm depths. However, the variability of these products tended to be too high at the month-to-month scale (i.e., high-frequency, short-period end of the spectrum), and too low at the decadal scale (i.e., low-frequency, long-period end of the spectrum) at the 50−100 cm depth (Figures 6 and A12).

The long-term trends in the regionally averaged SM of the merged products showed ranges that were centered around similar values as the source datasets, and were within the ranges of the source datasets except for a few cases (EC ORS 10−30 cm Sahara and West Asia, 30−50 cm Sahara) (Figure A13). For both the merged products and the source datasets, the most negative trends occurred in northeast Brazil for 0−10, 10−30, and 30−50 cm, and in south Australia/New Zealand for 50−100 cm, and the most positive values occurred in Alaska/northwest Canada, southeast South America, and north Asia (Figure A13). Therefore, the merging procedure was unlikely to have caused underestimation of the trends of the merged products. The occasional underestimations by the EC ORS dataset might be caused by uncertainty in the precipitation and temperature trends in arid regions. In the Sahara and West Asian regions, the CRU TS v4.03 dataset, which was used as observational constraint in the EC procedure (Sect. 2.4), had either more positive or more negative trends in precipitation than the drivers of the source datasets, and mostly less positive trends in temperature. Since temperature was positively correlated with the 10−30 cm and 30−50 cm SM in the EC relationship in the Sahara and West Asian regions (Figure A3), the negative bias in temperature trends would be consistent with the negative bias in SM trends.

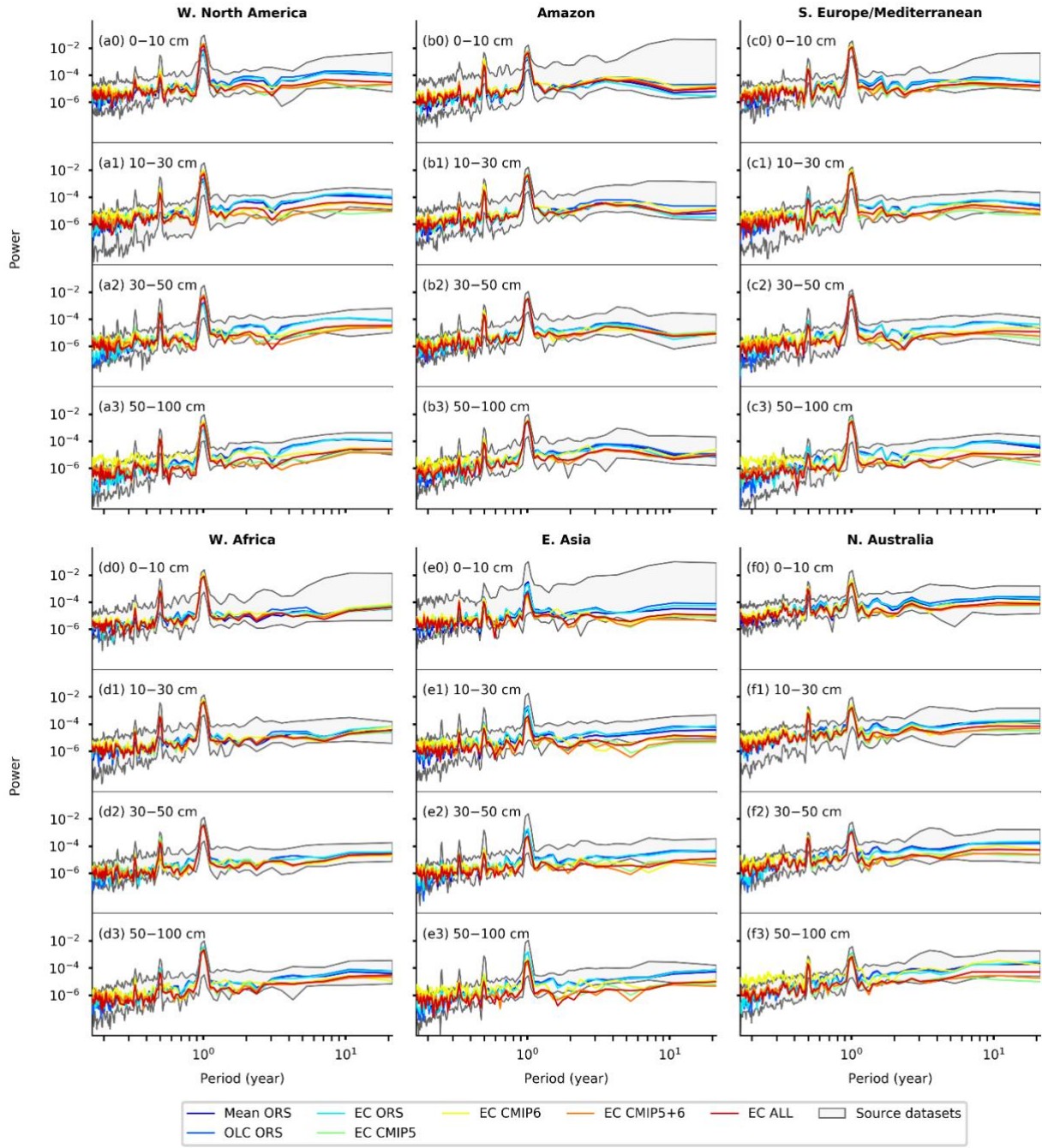

**Figure 6: The power spectral density of the spatially averaged time series of monthly SM over selected IPCC SREX regions (Field et al., 2012). The power spectral densities of the source datasets (ORS, CMIP5, and CMIP6) were calculated for each individual source dataset, and the displayed envelopes encompass the minimum to maximum ranges. Abbreviations: W. – west, N. – north, S. – south, E. – east.**

490

The SM climatology showed reasonable spatial patterns in all the merged products, with the lowest values occurring in the arid regions of Sahara, western United States, central Asia, and interior Australia, and highest values occurring in the high latitudes and tropical rainforest regions (Figure A15). The OLC merging method caused an increase in absolute SM values, especially in the 0–10 and 10–30 cm soil layers, relative to unweighted averaging (Figure A14, 1st and 2nd rows). The EC method did not induce a similar increase (Figure A15, 1st and 3rd rows), which was expected because the procedure did not change the 1981–2010 climatology of the source datasets (Sect. 2.8). The Mean ORS, EC ORS, EC CMIP5+6, and EC ALL products showed little difference in SM climatology across the soil layers, but the OLC ORS and EC CMIP6 products showed decreased SM from the shallower to deeper soil layers (Figure A15). The EC CMIP5 had the highest SM values at the 30–50 cm soil layer (Figure A15).

The timings of annual maximum SM were mostly consistent across different merged products, with exceptions occurring in northeast Asia, eastern Canada, and Alaska (Figure A16). The maximum SM occurred around February in the southern subtropics, southern North America, and the Mediterranean; around September in the monsoonal regions of Africa and southern and eastern Asia; and around May in northern North America and most of Eurasia. At deeper soil layers (30–50 and 50–100 cm), the CMIP5- and CMIP6-based merged SM showed earlier occurrence of annual maximum SM (around June) than the other merged datasets (around September) in eastern Asia.

All the merged products showed increasing SM trends in the northern high-latitudes, central Eurasia, and northern Africa, and decreasing trends in eastern South America, southern Africa, and eastern Australia (Figure A17). The CMIP5- and CMIP6-based merged datasets showed greater drying in eastern North America and Europe than the ORS-based estimates, and less drying near the North China Plain than the ORS-based products. A major difference existed between the CMIP6-based merged products (EC CMIP6, EC CMIP5+6, EC ALL) and the other products in northeast Asia in the 50–100 cm soil layer, where the former displayed strong drying trends and the latter did not. The estimated uncertainty intervals of the merged products were slightly larger for the OLC method than the unweighted standard deviation of the source datasets, and both were considerably larger than the EC method (Figure A18). For all the methods, the uncertainty intervals were greater in the temperate regions than in the arid regions, which was consistent with the higher SM absolute values in the temperature regions.

### 3.6 Sensitivity to precipitation, air temperature, and surface downwelling shortwave radiation

Based on partial correlation, precipitation was the dominant control of SM variability in the ORS-based products and in EC ALL over most of the globe (Figure 7), and generally had significant positive partial correlations (Figure A19). Precipitation was also the dominant control of SM variability in the CMIP5- and CMIP6-based products in the 0−10 and 10−30 cm layers, but not in the 30−50 and 50−100 cm layers (Figure 7), where the partial correlations between precipitation and SM were insignificant across most of the global land surface (Figure A18). In all the merged products, air temperature had significant negative partial correlations with SM in southwestern United States, eastern South America, southern Africa, the Mediterranean, and Australia (Figure A20). Some significantly positive correlations between temperature and SM existed in the Sahara, central Asia, and Tibetan Plateau regions (Figure A20). The primarily negative correlations were consistent with

the physical expectation that higher temperatures induce higher evaporative demand and thus lower SM. The CMIP5- and CMIP6-based products had stronger negative correlations between temperature and SM than the ORS-based products in Europe, which may explain the former products' more negative trends in SM in this region (Figure A17). Downwelling shortwave radiation was rarely a dominant control of SM variability in the ORS-based products (Figure 7). For the CMIP5- and CMIP6-based products, downwelling shortwave radiation was only a dominant control of SM variability in some of the mid-to-high-latitude and tropical rainforest regions (Figure 7), which were consistent with the distribution of light-limited ecosystems.

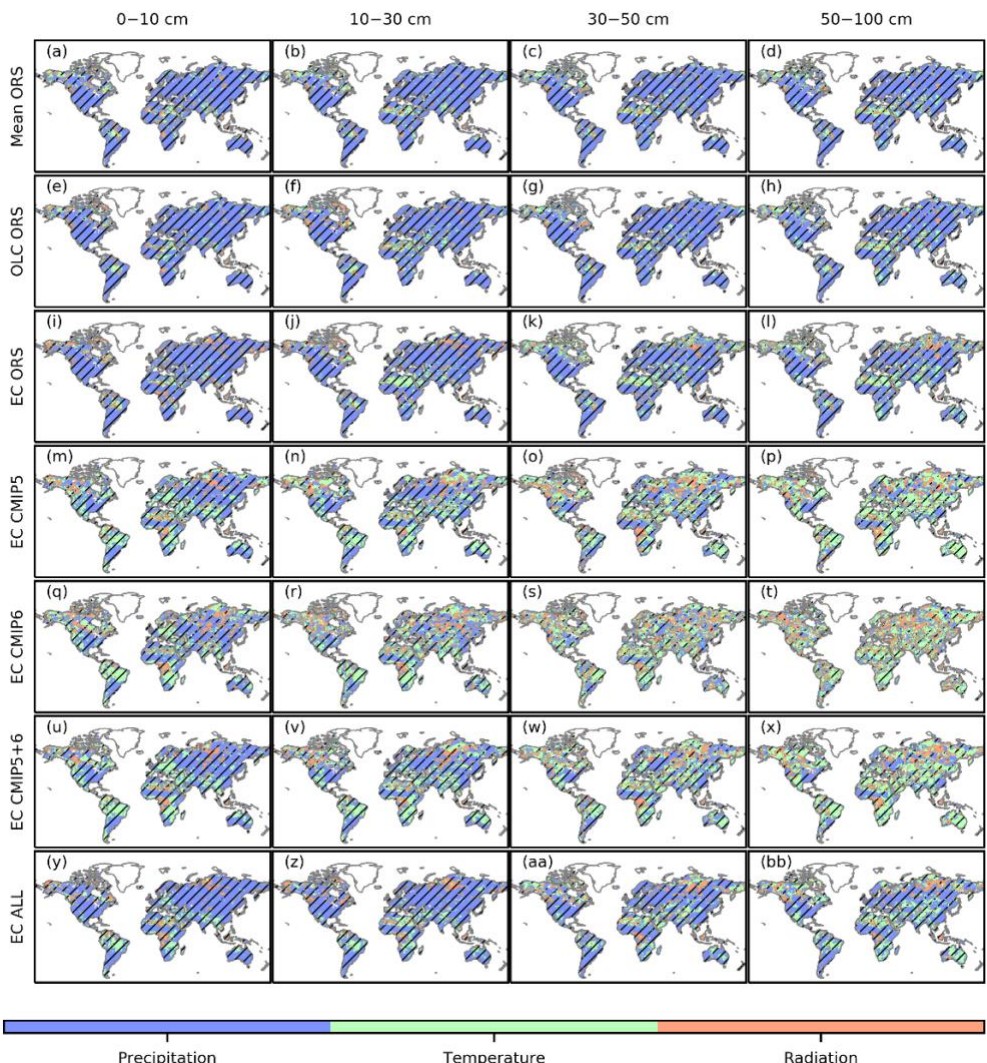

**Figure 7: The variable that has the best explanatory power for the inter-annual variability in SM in each grid for the different merged products and depths. The best explanatory power was defined as having the highest absolute partial correlation in the partial correlation analysis between annual mean SM and the annual mean meteorological variables. Hatching indicates that the partial correlation of the best-explanatory variable was significant at $p = 0.05$.**

## 4 Discussion

Overall, the merged SM products showed better performances than their source datasets (Sect. 3.1 and 3.2), temporal homogeneity (Sect. 3.3), the ability to capture large-scale drought events (Sect. 3.4), reasonable spatiotemporal patterns (Sect. 3.5), and reasonable climatic response characteristics (Sect. 3.6) across the globe and multiple soil layers. The ranges of performance metrics of the new datasets against in situ data (Figures 2 and A5) were broadly within the estimates reported by previous SM evaluations, although making a strict comparison is difficult because of the widely different spatiotemporal coverages and resolutions (Beck et al., 2021; Karthikeyan et al., 2017; Li et al., 2020b; Wang et al., 2021a; Yuan and Quiring, 2017). These results demonstrated that the merging procedures (unweighted averaging, OLC, EC) used were effective in creating relatively accurate long-term multi-layer SM data at the global scale.

Regarding the three merging methods, the OLC method only showed better performance than unweighted averaging over the urban and built-up lands, crop/natural vegetation mosaic, and barren land cover types (Figure A5), which may be a benefit rendered by the overrepresentation of these land cover types in the in situ observations (Figure A1). The ISMN stations are very sparse (Figure A1), and a previous study suggested that denser observations may lead to better-performing merged product (Gruber et al., 2018). In the future, data sources such as FLUXNET and local SM networks that are not included in the ISMN may be exploited to improve the OLC-ORS product. Future extension of the OLC method may aim to account for the spatial representativeness of individual stations (Molero et al., 2018), and to test alternative error-estimation methods such as the extended collocation (Gruber et al., 2016). The EC method showed similar performance to the unweighted averaging when applied onto the ORS source datasets, which may be because the meteorological forcings for these datasets were already realistic (Table A5). However, the effectiveness of the EC method was clear when applied onto the online CMIP5 and CMIP6 simulations. Despite such EC-based improvement, the ORS-based merged products tended to perform better than the EC CMIP5, CMIP6, CMIP5+6, and ALL products (Figures 2 and A5). The current EC procedure used simple linear regression form and only temperature and precipitation as constraint variables (Sect. 2.8). In addition to temperature and precipitation, the SM was influenced by other atmospheric and land conditions (e.g., wind, leaf area and stomata closure, snow cover and melt, groundwater). Therefore, future studies may achieve better EC outcomes by incorporating more influencing factors into the EC procedure, and by using nonlinear regression methods such as machine learning. Another drawback of the current EC method is the low uncertainty interval (Figure A18), which is likely an underestimation. Whereas the OLC method accounts for the difference between in situ and source datasets in estimating the uncertainty interval (Sect. 2.7), the EC method does not. Future studies should also aim to better incorporate this information into the estimation of EC-based uncertainty interval, and to better account for the structural uncertainty introduced by the regression form and limited range of predictors in the EC procedure.

The high performance variability of merged products across space (Figure A5) is consistent with previous studies (Beck et al., 2021; Karthikeyan et al., 2017; Li et al., 2020b; Wang et al., 2021a; Yuan and Quiring, 2017). The high RMSE of the merged products in the shallower soil layers across the water bodies and evergreen needleleaf forests (Figure A5e–h) were likely

caused by the high positive Bias in these land cover types (Figure A5a–d) since the corresponding Corr values were relatively high (Figure A5i–l). The positive Bias over water bodies may be caused by inaccurate land-water classification at the resolution of the source datasets (Tables A1–A4). The positive and negative Bias over the forested land cover types in high-latitude and tropical regions (e.g., evergreen needleleaf forests, evergreen broadleaf forests, and deciduous needleleaf forests; Figure A2) may be due to biases in evapotranspiration and leaf area index in the source LSMs, reanalysis, and ESMs (Tables A2–A4), and further related to processes such as rooting depth and hydraulic redistribution (Pan et al., 2020; Wang et al., 2021b). Low Corr occurred over some land cover types in high latitudes, semi-arid to arid regions, and urban areas (e.g., open shrublands, urban and built-up lands, cropland/natural vegetation mosaics, and barren lands; Figure A2). In the high latitudes, the low Corr may be associated with inadequate frozen soil processes in the source LSMs, reanalysis, and ESMs (Andresen et al., 2020). In the semi-arid to arid regions, the low Corr may be due to random errors in SM observations and simulated values, which would be comparatively large for low SM values. In urban areas, the low performance may be caused by the radio frequency interference of satellite observations (Wang et al., 2012) and inadequacies in the representation of urban areas at the resolution of the source model products (Table A2–Table A4).

When evaluated against the global and regional gridded datasets that were not used in the merging, the merged products showed the highest RMSE in climatology and lowest Corr in spatial trends (Figure 3; see the blue numbers above each panel). Such results were likely because the climatology SM values had higher magnitudes than the seasonal and interannual anomalies or trends, and the historical SM trends were highly uncertain. The low similarity between the SMOS L3/L4 and the synthesized SM products may be caused by the short overlapping period (2010–2016, Table 1). A previous study also found systematic differences between the climatology of satellite-observed and simulated SM (Piles et al., 2019). The high similarity between the SoMo.ml and GLEAM v3.3a root zone SM datasets and the merged products may be because the former depend on the same ISMN stations, from which the OLC-ORS product was derived (Sect. 2.2), and the latter were from the same reanalysis as the GLEAM v3.3a surface dataset used in the merging (Table A2). In general, because these evaluation datasets are not ground truths like in situ observations, the identified differences in evaluation metrics should not be viewed as an absolute indicator of unreliability in the merged products. Similarly, the benchmarking against scPDSI (Sect. 3.3) only provided qualitative rather than quantitative indicators of performance for the merged products because scPDSI is essentially a different variable from SM.

The vertical gradient in Bias (Figure 2, Figure A5), the high uncertainty in the vertical gradient in the climatology of merged SM (Figure A15), and the divergent trends in the 50–100 cm SM in northeast Asia across the merged products (Figure A17) point to the need to reduce uncertainties in vertical distribution and dynamics of SM in the merged products. The high SM values for the HadGEM2-CC and HadGEM-ES datasets may be the reason why the highest SM occurred in the 30–50 cm layer in the EC CMIP5 product (Figure A22). All the source datasets for the EC CMIP6 SM showed negative trends in northeast Asia in the 50–100 cm soil layer, but this feature does not exist in the original ORS or CMIP5 datasets (results not shown). All the source datasets do not have consistent SM vertical gradients, with the maximum value falling at either the surface, deepest,

or intermediate soil layers (Figure A22). Such vertical inconsistencies may be related to inconsistencies in the vertical discretization of the soil column (Tables A2–A4), soil properties in each layer, modeling of lateral flow and drainage, or other factors (e.g., Balsamo et al., 2009; Best et al., 2011; Melton et al., 2019). Previous regional or global SM evaluations (e.g., Beck et al., 2021; Karthikeyan et al., 2017; Li et al., 2020b; Wang et al., 2021a; Yuan and Quiring, 2017) rarely focused on the performance on vertical gradient, and such a limitation should be better addressed in future analyses and dataset development.

The temporal homogeneity test showed that discontinuity in variance existed in all the merged products, which may arise from several sources. The reference datasets, SMERGE v2 and GLEAM v3.3a 0−100 cm, were not perfectly homogeneous because both datasets assimilate satellite observations made by different instruments over time (Tobin et al., 2019; Martens et al., 2017). The observation systems assimilated by the reanalysis datasets of the ORS (Table A2) also change over time, thereby leading to potential discontinuities in the ORS-based products. Such limitations cannot be eliminated, considering the paucity of records before the satellite era, and the continuous evolution of observational and reanalysis systems. Discontinuity in a statistical sense can also be caused by changes in land use and other types of disturbances between two time periods, but such apparent discontinuity reflects real world situations.

The SM seasonality in the merged products (Figure A16) was broadly consistent with previously reported timing of annual maximum precipitation (Knoben et al., 2019). Differences at the deeper soil layers between the CMIP5- and CMIP6-based merged products and the ORS-based products may be partially caused by the lack of consideration of lagged SM response to meteorological drivers, especially at the deeper layers, in the EC method (Sect. 2.8). Uncertainty at the deeper layers would also be high because fewer source datasets were available than for the shallower layers (Tables A1–A4). The SM trends (Figure A17) were broadly consistent with previous reports on historical changes in agricultural droughts (Dai and Zhao, 2017; Liu et al., 2019; Lu et al., 2019).

The primarily positive partial correlations between the SM and precipitation, and the primarily negative partial correlations between the SM and air temperature or shortwave radiation, were consistent with expectations from physical processes. The existence of significant positive partial correlations between air temperature and SM might be caused by less precipitation falling as snow at higher temperatures in the cold Tibetan Plateau, and might be caused by stronger land-atmosphere feedbacks at higher temperatures in the Sahara and central Asia. The weaker relationships between precipitation and SM in the CMIP5- and CMIP6-based merged products than these in the ORS-based products (Figure A15) may be because the EC method did not fully explain the temporal mismatch between the source datasets and the real world because the relationships were insignificant in various grids and time steps (Figure A4). The stronger relationships between air temperature and SM of the CMIP5- and CMIP6-based merged products than these of the ORS-based products (Figure A16) may be partially caused by compensation for the weaker relationships between shortwave radiation and SM (Figure A17). Because the temperature and shortwave radiation tend to be highly correlated, shortwave radiation was not considered as a predictor for the EC method in the present research.

## 5 Data availability

The seven SM products, including the estimated SM values and uncertainty intervals, are available from https://doi.org/10.6084/m9.figshare.13661312.v1 (Wang and Mao, 2021). The files are in NetCDF4 format.

## 6 Code availability

The source codes for developing all the SM datasets are available at
https://ywang11@bitbucket.org/ywang11/soil_moisture_merge.git

## 7 Conclusions

This study achieved the goal of creating long-term, gap-free, multi-layer SM products (1970–2016, 0.5°, monthly, 0–10, 10–30, 30–50, and 50–100 cm) that displayed realistic temporal evolutions and spatial patterns and outperformed the source SM datasets in the systematic evaluations against independent in situ measurements and semi-independent gridded SM estimates. Three new SM products (mean ORS, OLC ORS, and EC ORS) developed from the satellite observations, reanalysis, and 645 offline LSMs were shown to perform better than those based on the ESMs. Therefore, they are recommended for future applications, such as the detection and attribution of historical changes of SM and associated extreme events, providing the initial and boundary conditions for atmospheric models, benchmarking various types of models, and managing drought and flood risks. By comparing three different merging methods (unweighted averaging, OLC, and EC), this study further denoted that the OLC method may require more in situ observations to exceed the unweighted averaging, and that linear regression-650 based EC with a limited range of un-lagged predictors was inadequate in correcting all the ESM errors. Future SM developments may aim to assemble more in situ SM datasets and to implement other advanced fusion algorithms (e.g., extended collocation, machine learning).

## 8 Author contributions

J.M. conceived the research; Y.W. and J.M. performed the analyses and drafted the figures; Y.W. and J.M. wrote the first draft 655 of the manuscript; M.J., F.M.H., X.S., S.D.W., and Y.D. reviewed and edited the manuscript before submission. All authors made substantial contributions to the discussion of content.

## 9 Competing interests

The authors declare that they have no conflict of interest.

**Acknowledgements**

This research was supported by an Oak Ridge National Laboratory (ORNL) subcontract (4000169153), and the Reducing Uncertainties in Biogeochemical Interactions through Synthesis and Computation Science Focus Area (RUBISCO SFA) project in the Earth and Environmental Systems Sciences Division (EESSD) of the Biological and Environmental Research (BER) office in the US Department of Energy (DOE) Office of Science. ORNL is managed by UT-BATTELLE, LLC, for DOE under Contract No. DE-AC05-00OR22725. The Dai self-calibrated Palmer Drought Severity Index data was provided

by the NOAA/OAR/ESRL PSL, Boulder, Colorado, USA, from their website at https://psl.noaa.gov/. The GSWP3 meteorological data was downloaded as the prepared historical input datasets for ISIMIP3a. For their roles in producing, coordinating, and making available the ISIMIP input data and impact model output, we acknowledge the modeling groups, the ISIMIP sector coordinators and the ISIMIP cross-sectoral science team.

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
