# Peer review of "Development of Observation-based Global Multi-layer Soil Moisture Products for 1970 to 2016"

_Earth System Science Data, 2021_

## Author Response (AR1)

**Reviewer #1**

By synthesizing a wide range of SM datasets using statistical methods, this study developed and evaluated seven global, long-term, multi-layer monthly SM products, which are crucial for many research fields. I appreciate the great efforts made by the authors and I believe that this study will definitely contribute to better understanding the global water, energy and biogeochemical cycles. Generally, the topic is very interesting and the material was well organized. However, some clarifications are needed to improve the quality of the manuscript, and I'd like to recommend an acceptance of the manuscript for publication after minor revisions.

**Response:** Thank you for the encouragement.

Comments and suggestions:

1. Section 2.1, three time ranges have been used in OLC method, please give necessary explanation on this.

**Response:** Thank you for the comment. An explanation is now provided in Section 2.1 in lines 105–107 (untracked version; tracked version lines 113–116):

"For the OLC method, the ORS datasets were grouped based on three time ranges (1970–2010, 1981–2010, and 1981–2016) that were selected to maximize the available ORS datasets in each time range."

2. Section 2.2, please describe the details about the usage of the ISMN observation to train the OLC method.

**Response:** Thank you for the comment. We added some descriptions and a reference to Sect. 2.7, where the OLC method was described fully, in lines 142–144 (untracked version; tracked version lines 151−154):

"After the ISMN observations were aggregated to monthly 0.5° resolutions, 60% of the month-grids were used as the observed SM values in the OLC method (the $o^{tj}$ variable; see Sect. 2.7), and the remainder were reserved for evaluating all the merged products."

3. Figure 3, what is the purpose of using SMOS L3, SoMo, SMVERGE v2,SOMS L4 and GLEAM v3.3a in the analysis? The relevant discussion is not clear enough.

**Response:** Thank you for your question. We have recognized that the original description for why and how these datasets were used was too brief. We expanded Sect. 2.5 to give a more detailed description of the rationale for using these datasets in lines 188–196 (untracked version; tracked version lines 197−206):

"A recent discussion on the evaluation of coarse-scale soil moisture datasets noted that neither in situ observations, which have limited coverage and small spatial footprint, nor satellite and LSMs, which have retrieval or modeling errors, can be considered fully adequate for evaluation

at the global scale; as such, a sound evaluation practice would require combining multiple sources of data (Gruber et al. 2020). Following this recommendation, the merged SM products were evaluated against the reserved 40% in situ observations (Sect. 2.2), as well as a few gridded reanalysis, satellite, and machine learning–upscaled SM datasets. Although the merging process aimed to use as many existing SM datasets as possible, the gridded datasets in Table 1 were not used in the merging because of incompatible vertical resolution, non-global spatial coverage, or short temporal coverage (Table 1). Such evaluation against multi-source gridded datasets complements the evaluation against in situ observations by providing sanity checks on the behavior of the merged products at large scales."

We also provided a more detailed description of how the evaluation was performed using the SMOS L3, SoMo, SMVERGE v2, SOMS L4, and GLEAM v3.3a datasets in Sect. 2.5 in lines 211–224 (untracked version; tracked version lines 221–234):

"The merged products were evaluated against the validation set of in situ observations and the gridded SM datasets using three common metrics: mean bias (Bias), root mean squared error (RMSE), and Pearson correlation coefficient (Corr). For evaluation against the in situ observations, the metrics were calculated both for the whole validation set and for each land cover type in consideration of the uneven distribution of ISMN observations across land cover types (Figure A1). The observational values used in each calculation were the land-cover–weighted averages (see Sect. 2.2), and the merged values were from the grids and time steps that have the observational values. For evaluation against the SMOS L3 gridded dataset, the 0–10 cm layer of the merged products and the source datasets (ORS, CMIP5, and CMIP6) were used. For evaluation against the other evaluation datasets, the merged and source datasets were linearly interpolated to depths of the evaluation datasets. The annual climatology, mean seasonal anomalies (i.e., the climatology of individual months minus the annual climatology), least-squares linear trends, and anomalies (i.e., the original values minus the mean seasonal cycle and trends) were calculated for each common grid cell and over the common time period between each pair of evaluated and evaluation datasets. Then, for each characteristic (climatology, seasonal cycle, linear trends, or anomalies), the Bias, RMSE, and Corr were calculated using the values of the characteristic pooled over all the common grid cells. When calculating the Bias, RMSE, and Corr for the trends, the insignificant trends at $p = 0.1$ were set to zero to prevent small random variability from influencing the results."

4. Figure 4, Mean ORS, OLC ORS and EC ORS show large bias after 1988, however, they give high correlations. Why?

**Response:** Thank you for pointing this out. Correlations and bias do not depict the same aspect of errors and can be unrelated to each other. In Figure 5 (originally Figure 4), we used the Spearman rank correlation, which is Pearson correlation between ranks. That is, for a pair of time series, $x_i$ and $y_i$ ($i = 1, 2, \ldots, N$), the calculation procedure converts each $x_i$ into its rank among all the $x$-values, and each $y_i$ into its rank among all the $y$-values, and then calculates the Pearson correlation between the ranks. Therefore, the Spearman correlation is not sensitive to the original magnitudes of the $x$- and $y$-series. We chose the Spearman correlation because scPDSI is a normalized index, and its magnitude is not comparable to the magnitude of soil moisture. To clarify the rationale of this choice and its consequence, we added the following text to the

description of Sect. 3.4 (originally Sect. 3.3), in lines 441−446 (untracked version; tracked version 443−477):

"To better quantify the similarity between the scPDSI and SM anomalies, Spearman correlations (Hollander et al., 2013) were calculated and are shown above each panel in Figures 4 and A6−A8. The Spearman correlation metric was deemed suitable for measuring the similarity because the magnitudes of scPDSI, which is a unitless standardized index, and of SM anomalies ($m^3/m^3$), are not comparable. Spearman correlation is not sensitive to magnitudes because the metric is calculated using the rank of each *x*-value among all the *x*-values, and the rank of each *y*-value among all the *y*-values, for an *x*-*y* pair of time series (Hollander et al., 2013)."

We also generally revised Sect. 3.4 (originally Sect. 3.3) to describe Figures 5 (originally Figure 4) and A6–A8 (originally A6–A8) in more detail and clarity.

5. Figure 5, too many plots present limited useful information, which should be improved.

**Response:** Thank you for the suggestion. The original intention of Figure 5 was to show that the time series of the merged products did not have temporal discontinuity and had reasonable temporal variability. In recognition of this comment, and the fact that time series plots did not provide quantifiable measures on temporal discontinuity and variability, we added a homogeneity test of whether temporal discontinuity existed in the concatenated datasets (OLC ORS, EC ORS, and EC ALL), replaced Figure 5 with a spectrum analysis, and reduced the number of panels shown in the main text.

The method of the added homogeneity test is described at the end of Sect. 2.9, lines 336−352 (untracked version; tracked version lines 346−362), and the results of the homogeneity test are described in the new Sect. 3.3, lines 407−425 (untracked version; tracked version lines 435−460). In summary, no discontinuities in mean were identified in the merged products. Some discontinuities in variance were identified, but the concatenated datasets (OLC ORS, EC ORS, and EC ALL), which used a few different sets of source datasets for different time periods, were not found to be more discontinuous than the non-concatenated datasets (Mean ORS, EC CMIP5, EC CMIP6, EC CMIP5+6), which used the same source datasets throughout. This result demonstrated that the concatenation practice did not introduce additional discontinuities into the merged data. Discussion about the potential causes of discontinuity in the merged products are added to lines 598−605.

The new spectrum analysis is shown in Figure 6 (originally Figure 5), and the description of Figure 6 in Sect. 3.5 (originally Sect. 3.4), lines 459−476 (untracked version; tracked version 436−454), was re-rewritten to reflect the new figure. The power spectral densities of the ORS-based merged products generally fall within the envelopes of the source datasets. The same was true for the CMIP5- and CMIP6-based merged products, except at the 50–100 cm depth. These results demonstrated that the ORS-based products were better than the CMIP5- and CMIP6-based merged products, and had reasonable temporal variability.

A few lines were added to the abstract to reflect the added homogeneity test and spectrum analysis, lines 25−28 (untracked version; tracked version lines 27−31):

"The merged products generally showed reasonable temporal homogeneity and physically plausible global sensitivities to observed meteorological factors, except that the ESM-dependent products underestimated the low-frequency temporal variability in SM, and over-estimated the high-frequency variability for the 50–100 cm depth."

6. Section 3.5, the discussion is not deep enough, which provides limited information to the readers.

**Response:** Thank you for the comment. The original figure accompanying the original Sect. 3.5 (new Sect. 3.6) shows globally aggregated probability density distributions, which has limited information content that precluded detailed discussion. We replaced it with a summary of the dominant control on soil moisture at the grid scale (Figure 7, originally Figure 6), and rewrote the entire section (untracked version 504−521; tracked version 548−582). The results showed that precipitation was the dominant control of soil moisture and had generally significant positive correlations with soil moisture. Temperature was a more important control of soil moisture in the CMIP5- and CMIP6-based datasets than in the ORS-based datasets, but in all the merged datasets, any significant correlations between temperature and soil moisture tended to be negative. Solar radiation was an important control of soil moisture only in the CMIP5- and CMIP6-based datasets, in the northern mid-to-high latitudes and tropical forests, which might be caused by light limitation on ecosystems. These results are plausible compared to physical mechanisms and show that the merged products had reasonable sensitivities to meteorological forcings.

Although the original Figure 6 showed seasonal variations in partial correlations, we chose to no longer show the seasonal variations in the new Figure 7, because they will result in too many graphs and are not of any particular importance to the evaluation.

**Reviewer #2**

Global soil moisture products were developed based on different combinations of existing soil moisture datasets and methods in this study. These products are gap-free, long-term, and multi-layer. I think the developed soil moisture products will be important given the lack of global and consistent soil moisture observations. The soil moisture datasets used to create the products were based on different techniques (e.g., remote sensing, modeling) and different specifications (e.g., soil depth, time step). Merging them together is a challenging task.

-The manuscript highlighted that the products are gap-free, long-term and multi-layer, but it seems the verification didn't assess these advancements. For example, compare to the existing datasets, how the products perform when/where gaps exist, and how the products perform in different layers.

**Response**: Thank you for the helpful comments and questions.

Among the existing gridded soil moisture datasets, only the satellite datasets had gaps. These gaps are not the same across satellites or across different time steps for the same satellite, and therefore were not assessed specifically in the study. The existing modeled and reanalysis

datasets are already gap-free, and some have longer temporal coverage than the current dataset, but they have larger errors than the merged products, which can be seen in Figure 2. Therefore, the contribution of this study is more to reduce the errors of the existing gap-free, long-term, multi-layer datasets, rather than filling gaps.

To more accurately describe the characteristics of existing datasets, we revised the introduction to read as follows in lines 44−47 (untracked version; tracked version lines 49−51):

"The SM in LSM simulations usually spans multiple soil layers, and has no spatial or temporal gaps, which is convenient for regional and global analysis (Gu et al., 2019); however, LSM simulations may contain considerable errors because of inadequacies in the model physics, parameterization, and drivers (Andresen et al., 2020)."

The performances of the merged products in different layers were already presented. In Figure 2, Figure 6 (originally Figure 5), and Figure 7 (originally Figure 6), the evaluation results for 0–10, 10–30, 30–50, and 50–100 cm were presented individually. In Figure 3, the gridded soil moisture datasets for evaluation also covered different layer depths. The evaluation against the self-calibrated Palmer Drought Severity Index was also made separately for 0–10 cm (Figure 5, originally Figure 4; Figure A6) and 0–100 cm (Figures A7 and A8). To highlight these assessments, we revised the first sentence of the Sect. 4, lines 527−529 (untracked version; tracked version lines 584−587), as follows:

"Overall, the merged SM products showed better performances than their source datasets (Sect. 3.1 and 3.2), temporal homogeneity (Sect. 3.3), the ability to capture large-scale drought events (Sect. 3.4), reasonable spatiotemporal patterns (Sect. 3.5), and reasonable climatic response characteristics (Sect. 3.6) across the globe and multiple soil layers."

-Fig. 3 includes bias, RMSE, and Corr of different statistics (i.e. climatology, seasonal cycle, trend,…). I am not sure if they can use the same range of color ramp in the same column to show the situations of different statistics because the statistics are in different units and magnitudes. It is unclear how seasonality was calculated and how the bias, RMSE and Corr of seasonality and trends were estimated.

**Response**: Thank you for pointing this out. We revised Figure 3 to show normalized values in each column, with the maximum value in each column always being set to 100%. In this way, the relative performances of the merged and source datasets are no longer obscured by the large differences across the evaluation datasets. We also revised the text of Sect. 3.2 to highlight this normalization, in lines 389−391 (untracked version; tracked version lines 403−405):

"To emphasize the differences between the merged and the source datasets, rather than across the evaluation datasets, Figure 3 displays the evaluation metrics in normalized units, with the maximum value across the merged and source datasets of each matric set to 100%."

To clarify how seasonality was calculated, and how the bias, RMSE, and Corr were estimated, we added the following explanation to Sect. 2.5, lines 211−224 (untracked version; tracked version lines 221−234):

"The merged products were evaluated against the validation set of in situ observations and the gridded SM datasets using three common metrics: mean bias (Bias), root mean squared error (RMSE), and Pearson correlation coefficient (Corr). For evaluation against the in situ observations, the metrics were calculated both for the whole validation set and for each land cover type in consideration of the uneven distribution of ISMN observations across land cover types (Figure A1). The observational values used in each calculation were the land-cover–weighted averages (see Sect. 2.2), and the merged values were from the grids and time steps that have the observational values. For evaluation against the SMOS L3 gridded dataset, the 0–10 cm layer of the merged products and the source datasets (ORS, CMIP5, and CMIP6) were used. For evaluation against the other evaluation datasets, the merged and source datasets were linearly interpolated to depths of the evaluation datasets. The annual climatology, mean seasonal anomalies (i.e., the climatology of individual months minus the annual climatology), least-squares linear trends, and anomalies (i.e., the original values minus the mean seasonal cycle and trends) were calculated for each common grid cell and over the common time period between each pair of evaluated and evaluation datasets. Then, for each characteristic (climatology, seasonal cycle, linear trends, or anomalies), the Bias, RMSE, and Corr were calculated using the values of the characteristic pooled over all the common grid cells. When calculating the Bias, RMSE, and Corr for the trends, the insignificant trends at $p = 0.1$ were set to zero to prevent small random variability from influencing the results."

-The products were developed by merging the different soil moisture datasets. I wonder if this would offset the temporal variability and trend of time series. Comparing the temporal variability and trend of the merged products with individual datasets (e.g., ESA CCI, ERA5…) like Fig. 5 would be helpful. Fig. 5 currently only shows the time series of the merged products.

**Response**: Thank you for pointing out this possibility. We now performed a check on temporal variability by comparing the power spectral densities of the merged products with the ranges of power spectral densities of the source datasets at regional levels. The results are shown in Figures 6 and A10 (originally Figure 5). The associated text in Sect. 3.5 (originally Sect. 3.4) is also revised (lines 459−470 in the untracked version; lines 491−512 in the tracked version). The power spectral densities of the ORS-based merged products generally fall within the envelopes of the source datasets. The same was true for the CMIP5- and CMIP6-based merged products, except at the 50–100 cm depth. These results demonstrated that the ORS-based products were better than the CMIP5- and CMIP6-based merged products, and had reasonable temporal variability.

We also performed a check on the trends of the merged products by comparing them against the ranges of trends of the source datasets. The results are shown in Figure A13, and the description is provided in lines 471−476 (untracked version; tracked version lines 513−518). The merged products had slightly larger ranges of trends than the source datasets, but were centered around approximately the same value in each region, and closely followed the source datasets in terms of the region-to-region variations in the trends. These results demonstrated that the merged products had reasonable trends.

-Abstract, provide a brief explanation of the better performance of the hybrid products without Earth System Models (ESMs) than those with ESM.

**Response:** Thank you for this suggestion. We revised the abstract and the relevant sentences now read as follows, lines 19−30 (untracked version; tracked version lines 19−33):

"……The merged products outperformed their source datasets when evaluated with in situ observations (mean bias from −0.044 to 0.033 $m^3/m^3$, root mean squared errors from 0.076 to 0.104 $m^3/m^3$, Pearson correlations from 0.35 to 0.67) and multiple gridded datasets that did not enter merging because of insufficient spatial, temporal, or soil layer coverage. Three of the new SM products, which were produced by applying any of the three merging methods onto the source datasets excluding the ESMs, had lower bias and root mean square errors and higher correlations than the ESM-dependent merged products. The ESM-independent products also showed a better ability to capture historical large-scale drought events than the ESM-dependent products. The merged products generally showed reasonable temporal homogeneity and physically plausible global sensitivities to observed meteorological factors, except that the ESM-dependent products underestimated the low-frequency temporal variability in SM, and over-estimated the high-frequency variability for the 50–100 cm depth. Based on these evaluation results, the three ESM-independent products were finally recommended for future applications because of their better performances than the ESM-dependent ones……"

-P2 L40, elaborate "its spatial gaps remain unresolved".

 **Response**: Thank you for the comment. We revised the wording to clarify the meaning, lines 42−44 (untracked version; tracked version lines 46−47):

"Although a long-term (1979–present) concatenated SM dataset was developed by merging data from multiple satellites, the merged product did not fill the spatial gaps that existed in the source satellite datasets (Dorigo et al., 2012; EODC, 2021)."

-P2 L51-52, explain why Global Climate Models (GCMs) were not considered in the study and the differences in GCMs and ESMs.

**Response:** Thank you for the comment. In our understanding, ESM is a newer and broader term that includes models that consider biogeochemical feedbacks, human dimensions, and likely in the near future the deep earth processes, whereas GCM is an older term that refers to models that focuses on atmospheric-ocean processes. The state-of-the-art models participating in CMIP5 and CMIP6 are generally ESMs. Therefore, we decided to keep using ESMs in the paper.

-P3 L66, explain why the merged products would likely perform better.

**Response**: Thank you for the suggestion. We revised the sentence to read as follows, lines 69−72 (untracked version; tracked version lines 74−76):

"Because of the incorporation of various quality-controlled observations in the merging process, the merged products would likely perform better than the SM in the original LSMs or ESMs while being gap-free in space and having long temporal and multi–soil-layer coverage."

-P3 L71, why in situ observations are not included in the unweighted averaging?

**Response:** Thank you for the question. We added a reference in line 75−76 (untracked version; tracked version 79−80; original L71) to Sect. 2.1:

"Unweighted averaging assigns equal weight to all the source datasets and does not use in situ information (see Sect. 2.1 for explanation for the exclusion)."

We also provided the full explanation in Sect. 2.1, lines 94−99 (untracked version; tracked version 99−104):

"The unweighted averaging did not use any in situ observations, because the in situ observations were sparse (~1,400 stations compared with ~60,000 grids in a 0.5° gap-free dataset over the global land surface; Sect. 2.2). In unweighted averaging, the in situ observations can only influence the merged values in the time steps and grids that coincided with the observations. Therefore, the inclusion of in situ observations would have little influence on the results of unweighted averaging. Also, to validate a merged time step and grid, an un-merged observation must be available at the same time step and grid, which would be difficult to achieve in data-sparse situations."

-P3 L78, it is unclear what are the variables that have/have no observations in ESMs

**Response:** Thank you for the question. It was a grammatical error. The sentence is now revised to read as follows, lines 81−84 (untracked version; tracked version 85−88):

"This method first uses data from multiple ESMs to establish physically meaningful and statistically significant relationship between the constraint variables that have observations and a target variable that has no observations, and then uses the relationship and actual observations to derive a constrained target variable (Mystakidis et al., 2016; Padrón et al., 2019)."

-P3 L90-93, this sentence is unclear. Please rephrase it. Observations of soil moisture/meteorological variables were used in some cases, but not in another. It is confusing in what cases observations were used. This is not presented in Fig. 1.

**Response**: Thank you for the comment. We revised Sect. 2.1 in response to this comment as well as the comment above on P3 L71. Sect. 2.1 now explains what observations were used with which method, lines 94−102 (untracked version; tracked version 99−109):

"The unweighted averaging did not use any in situ observations, because … to achieve in data-sparse situations. The OLC method used in situ observations to constrain the ORS datasets. The EC method (Mystakidis et al., 2016) was applied over the ORS, CMIP5, CMIP6, the combination of CMIP5 and CMIP6 (CMIP5+6), and the combination of ORS, CMIP5, and CMIP6 (ALL) datasets (Eyring et al., 2016; Taylor et al., 2012), and used gridded global meteorological observations as constraints."

-P4 L95-100, explain why ORS datasets were divided into different time ranges, and what "concatenation" means.

**Response**: Thank you for the comment. We added explanations, including a reference to Sect. 2.9, where the concatenation procedure was described, to Sect. 2.1, lines 106−111 (untracked version; tracked version lines 113−120):

"For the OLC method, the ORS datasets were grouped based on three time ranges (1970–2010, 1981–2010, and 1981–2016) that were selected to maximize the available ORS datasets in each time range. For each time range, the ORS datasets that fully cover the time range were merged with the OLC method; if an ORS dataset fully covers two or three time ranges, it was used in all the covered time ranges (see the "used time period" in Tables A1–A3). Then, the merged results for all three time ranges were concatenated into a consistent dataset covering the whole target period following a previous method for concatenating the remote-sensing SM (Dorigo et al., 2017; Liu et al., 2011, 2012) (Sect. 2.9)."

-P5 L123, "the number of observations that falls into each land cover type is displayed in Figure S2". Fig. S2 only shows land cover type.

**Response**: Thank you for the comment. The reference to Figure A2 was a mistake. The sentence has been revised to read as follows, in lines 136−137 (untracked version; tracked version 144−145):

"Figure A1 shows the aggregated ISMN observations at the 0.5° grid scale and the number of observations that falls into each land cover type."

-Table S1 and others, why did a dataset have different used time periods?

**Response**: Thank you for the question. We added clarification for Tables A1–A3 in the response to the above comment on P4 L95–100 as follows:

"if an ORS dataset fully covers two or three time ranges, it was used in all the covered time ranges (see the "used time period" in Tables A1–A3)"

-Section 3.3, it is interesting to compare SM anomalies with drought events. However, it seems the comparison is based on years instead of events. Furthermore, it is unclear how the drought events were selected and why only the two regions were considered.

**Response:** Thank you for the comment. The reason why the two drought events were selected from many prominent cases is now clarified in Sect. 2.6, lines 233−236 (untracked version; tracked version 242−245):

"The selected historical drought events were the United States drought of 1985–1992 and the Australian millennium drought of 2002–2009 because of their macro-regional spatial scale and high severity; many other drought events would also fit these criteria (Spinoni et al., 2019), but conducting a comprehensive assessment on drought events is beyond the scope of this study."

The figure shows years because the comparison was made on the spatial pattern in each year during these two multi-year events to show the progression and cessation of drought. We added a clarification in Sect. 2.6, lines 236−237 (untracked version; tracked version 245−246):

"A Self-Calibrated Palmer Drought Severity Index (scPDSI) dataset (Dai et al., 2004) was used as the benchmark, and the spatial patterns of SM anomalies and scPDSI were compared year-by-year during these two drought events."

We also modified the description in Sect. 3.4 (originally Sect. 3.3) to clarify the meaning of the graphs, lines 234−241 (untracked version; tracked version 462−470):

"Lower values in scPDSI and SM anomalies are indicative of drier conditions, and higher values indicate wetter conditions. For the United States 1985–1992 drought, the scPDSI, 0−10 cm SM anomalies, and 0−100 cm SM anomalies all showed gradual expansion of drought from 1985 to 1988, and gradual alleviation from 1989 to 1992, with the most severe drought being reached in the northern Great Plains in 1988 (Figures 4 and A7). For the Australia 2002–2009 drought, the ORS-based 0−10 and 0−100 cm SM anomalies captured the pan-Australian drought shown by the scPDSI in 2002−2003, 2005, and 2007−2009, and the eastern-Australian drought in 2004 and 2006 (Figures A6 and A8). The CMIP5- and CMIP6-based SM anomalies also mostly captured the Australian drought patterns but did not capture the pan-Australian drought in 2007 and 2008 (Figures A6 and A8)."

-Fig. 6, what is the y axis? The meanings of some lines are provided, e.g., dashed orange line.

**Response:** Thank you for the comment. The *y*-axis was the density value in a probability distribution, and the meaning of lines was meant to be a combination of the legends for line styles and colors (i.e., dashed orange line means DJF tas). But, we decided to replace Figure 7 (originally Figure 6) with a plot that shows the dominant control of the inter-annual variability in soil moisture, in consideration of Reviewer #1's comment about the lack of depth in the related discussion. The new figure more effectively summarizes the partial correlations shown in Figures A18–A20 (originally Figures A14–A16). The text of Sect. 3.6 (originally Sect. 3.5) was revised accordingly. The seasonal variations in partial correlations are not shown anymore because they would result in too many graphs and are not of any particular importance to the evaluation.

---

## Author Response (AR2)

Most of my comments have been addressed. I still have the following comments:

-Fig. 3 The meaning of the blue numbers at the top of each sub-figure is unclear.

**Response:** Thank you for the comment. The original explanation was in the caption and read as "The column-wise maximums to the precision of two decimal points are displayed above each column". We revised it to "The blue number at the top of each column is the column-wise maximum to the precision of two decimal points" to make the meaning clearer.

-Figs. 6 and A13 were supplemented to address my concern about the variations and trends of the merged products. However, the supplemented figures are not sufficient to answer my question. By merging different datasets, the averaging effect (a high value of a year in a product may be offset by a low value of that year in another product) may make the variations and trends smaller than individual datasets, which has been reported by previous studies. Moreover, Fig. A13 clearly shows that some merged products underestimated the trends compared to the range of individual datasets. A more precise discussion about this point is suggested.

**Response:** Thank you for raising the comment. We apologize for having made a mistake in Fig. A13. In the plotting script, an error made the full range of the source datasets not displayed. We double-checked the related script for Fig. 6 and did not find any mistake. The correct figure shows that the merged products were within the range of the source datasets except for the EC ORS dataset, in 10-30cm in the Sahara and West Asia regions, and in 30-50cm in the Sahara region. We suspect that the few underestimations were due to the uncertainty in precipitation and temperature in these arid regions, and revised Sect. 3.5 to discuss the issue (lines 475-488 in the tracked revised manuscript, lines 474-485 in the un-tracked revised manuscript)

We also revised the first sentence of Sect. 3.5 from "… the variability of the merged products may be damped…" to "… the variability and trends of the merged products may be damped…", to describe the phenomena more precisely.

- "Because of the incorporation of various quality-controlled observations in the merging process, the merged products would likely perform better than the SM in the original LSMs or ESMs while being gap-free in space and having long temporal and multi–soil-layer coverage." Being gap-free is not an advantage of the merged products over LSMs/ESMs, because LSMs/ESMs do not have gaps in their simulation outputs.

**Response:** Thank you for the comment. We revised this sentence to read "Because of the incorporation of various quality-controlled observations in the merging process, the merged products would likely perform better than the SM in the original LSMs or ESMs, while keeping the benefits of being gap-free in space and having long temporal and multi–soil-layer coverage". Hopefully this creates less confusion.